

# Adaptive Behavior of Over a Million Individual Farmers Under Consecutive Droughts: A Large-Scale Agent-Based Modeling Analysis in the Bhima Basin, India

Maurice W.M.L. Kalthof[1], Jens de Bruijn [1,2], Hans de Moel[1], Heidi Kreibich[3], Jeroen C.J.H[1]. Aerts

[1] Institute for Environmental Studies (IVM), Vrije Universiteit Amsterdam, Amsterdam, The Netherlands
[2] International Institute for Applied Systems Analysis (IIASA), Laxenburg, Austria
[3] Section Hydrology, GFZ German Research Centre for Geosciences, Potsdam, Germany

*Correspondence to:* Maurice W.M.L. Kalthof (w.m.l.kalthof@vu.nl)

**Abstract.** Consecutive droughts, becoming more likely, produce impacts beyond the sum of individual events by altering catchment hydrology and influencing farmers' adaptive responses. We use GEB, a coupled agent-based hydrological model, and expand it with the Subjective Expected Utility Theory (SEUT) to realistically simulate farmer behavior and subsequent hydrological interactions. We apply GEB to analyze the adaptive responses of ±1.4 million heterogeneous farmers in India's Bhima basin over consecutive droughts and compare scenarios with and without adaptation. In adaptive scenarios, farmers can either do nothing, switch crops, or dig wells, based on each action's expected utility. Our analysis examines how these adaptations affect profits, yields, and groundwater levels, considering, e.g., farm size, risk aversion and drought perception. Results indicate that farmers' adaptive responses can decrease drought vulnerability and impact after one drought (x6 yield loss reduction), but increase it over consecutive due to switching to water-intensive crops and homogeneous cultivation (+15% income drop). Moreover, adaptive patterns, vulnerability, and impacts vary spatiotemporally and between individuals. Lastly, ecological and social shocks can coincide to plummet farmer incomes. We recommend alternative or additional adaptations to wells to mitigate drought impact and emphasize the importance of coupled socio-hydrological ABMs for risk analysis or policy testing.

**Short summary.** Our study explores how farmers in India's Bhima basin respond to consecutive droughts. We simulated all farmers' individual choices—like changing crops or digging wells—and their effects on profits, yields, and water resources. Results show these adaptations, while improving incomes, ultimately increase drought vulnerability and damages. Such insights emphasize the need for alternative adaptations and highlight the value of socio-hydrology models in shaping policies to lessen drought impacts.

## 1 Introduction

Anthropogenic climate change and population growth has increased exposure of society to droughts (Smirnov et al., 2016). Furthermore, the growing demand on water is increasingly stressing fresh-water system, amplifying the impact of droughts (Best & Darby, 2020; van Loon et al., 2016). Therefore, there is a necessity to strive for drought risk adaptation both at larger scales by governments (e.g. reservoir management) and at the local scales by farmers through efficient water use and irrigation (UNDRR, 2015; Wilhite et al., 2014).

Empirical research into what factors drive adaptation is ongoing but mostly focuses on single events and at one point in time (Blauhut et al., 2016; P. D. Udmale et al., 2015). However, consecutive droughts are becoming more



likely and can result in impacts that differ from the sum of the individual events' parts (Anderegg et al., 2020; van
der Wiel et al., 2023; Zscheischler et al., 2020). Consecutive droughts impact farmer communities in a few distinct
(but interrelated-) processes. (1) The first (of consecutive) drought(s) can have a physical hydrological impact on
the second drought. For example, a lowered groundwater table after the first event may not have been replenished
before the second drought starts, which can limit the capacity for irrigation during the second drought (Anderegg
et al., 2020; van der Wiel et al., 2023; Zscheischler et al., 2020). (2) Moreover, socio-economic factors like income
or debts also influence the vulnerability of farmers and their ability to adapt during multiple drought events. For
example, the reduced income of farmers after a first drought (e.g. due to less yield) may lead to less financial
capacity to cope with the second drought. (3) Finally, behavioral factors such as risk aversion and risk perception
also play a role in how farmers adapt to (multiple-) droughts (Habiba et al., 2012; Ward et al., 2014). For example,
farmers can have an increased risk perception after the first event, which may lead to an accelerated
implementation of drought adaptation measures (Aerts et al., 2018; Habiba et al., 2012; Nelson et al., 2013; van
Duinen et al., 2015), thus reducing the impact of the second drought.
A key research challenge is to capture the spatial-temporal dynamic feedbacks between vulnerability, human
behavior and physical hydrological processes over periods with consecutive droughts (Cui et al., 2021; Trogrlić et
al., 2022; van der Wiel et al., 2023). Empirical data from surveys may support analysis about the factors driving
drought adaptation feedbacks. However, only few studies provide empirical data on the spatial-temporal drivers
of drought vulnerability and adaptation under multi-drought conditions (Kreibich et al., 2022). This is why current
drought risk assessment research suggests developing model-based approaches (Cui et al., 2021; Trogrlić et al.,
56  2022).

A special class of simulation models are agent-based models (ABMs). ABMs are specially designed to capture the
behavior of autonomous individuals (i.e. agents) (Blair & Buytaert, 2016; Schrieks et al., 2021; M. Wens et al.,
2019). When integrated with a hydrological model, they can also capture bi-directional human-water feedbacks,
with agents reacting to environmental changes (e.g., precipitation deficits) and impacting their surroundings (e.g.,
depleting groundwater levels) (De Bruijn et al., 2023). In contrast to other socio-hydrological models, ABMs can
simulate how drought adaptation of individual farmers is influenced by other agents. This is essential, as adaptive
feedbacks by farmers are heterogeneous and depend on the varying physical, socio-economic and behavioral
characteristics among the farmer population (e.g., risk aversion, income, farm size, adaptations,
upstream/downstream, proximity to reservoirs; Di Baldassarre et al., 2018; Habiba et al., 2012; P. Udmale et al.,
2014; P. D. Udmale et al., 2015). For example, government-led large-scale adaptation efforts, like reservoir
management, may affect farmers' irrigation usage (di Baldassarre et al., 2018). Additionally, agents can emulate
their neighbors' practices, such as cropping patterns (Baddeley, 2010). However, most ABM based studies that
simulate individual farmers remain at small scales (Zagaria et al., 2021), whereas studies at large basin scales
aggregate agents, data and processes and omit small scale behavior due to computational constraints (Castilla-Rho
et al., 2017; Hyun et al., 2019).
To address these challenges, De Bruijn et al. (2023) developed GEB, an ABM coupled with a hydrological model
(CWatM, Burek et al., 2020), that is able to model the behavior of millions of agents efficiently at one-to-one
scale. With GEB, it is possible to analyze the culminated hydrological and agricultural impacts of many small-
scale processes at river basin scale. However, to analyze the complex human decision-making process under
consecutive droughts we require behavior to change dynamically in response to drought events (Groeneveld et al.,



2017; Schrieks et al., 2021). In the current version of GEB this is not possible, as its decision rules for adaptation
are based on simple assumptions of human behavior (De Bruijn et al., 2023; Schrieks et al., 2021).
The main goal of this study is to assess the vulnerability and adaptive responses of farmer agents under consecutive
droughts. Therefore, we integrate the Subjective Expected Utility theory (SEUT, Fishburn, 1981) into the GEB
model. The SEUT is a well-established behavioral economic theory that explains farmer adaptation decisions as
economic maximization under risk, influenced by subjective factors such as risk aversion and perception. By
parametrizing and calibrating the SEUT with local data and letting the risk perception change dynamically in
response to drought events, we attempt to create a more accurate depiction of adaptation under consecutive
droughts. We further refine our characterization of farmers—including their drought experience, adaptation costs,
and loan debts—to better understand changes in their individual vulnerability and risk, such as fluctuations in
income, debt levels, adaptation uptake, and groundwater levels. We apply and calibrate the augmented GEB in the
Bhima basin, which is part of the Krishna basin in India. Our work helps in understanding how consecutive drought
events affect different types of farmer's vulnerability and impact. The paper is organized as follows: We begin
with a high-level overview of the model setup (2.1) and a description of the study area (2.2). We then detail our
implementation of behavior (2.3), crop cultivation methods (2.4), agent initialization (2.5), and conclude with
model calibration and scenario setup (2.6). Next, in the results section, we analyze the evolution of model
vulnerability and risk parameters over consecutive droughts in an adaptation scenario (3.1), compare it to a no-
adaptation scenario (3.2), and review the results of the sensitivity analysis (3.3). This leads into a discussion of our
key findings and challenges to our methods (4). Finally, we summarize our conclusions and suggest directions for
future research (5).
**2 Methods**

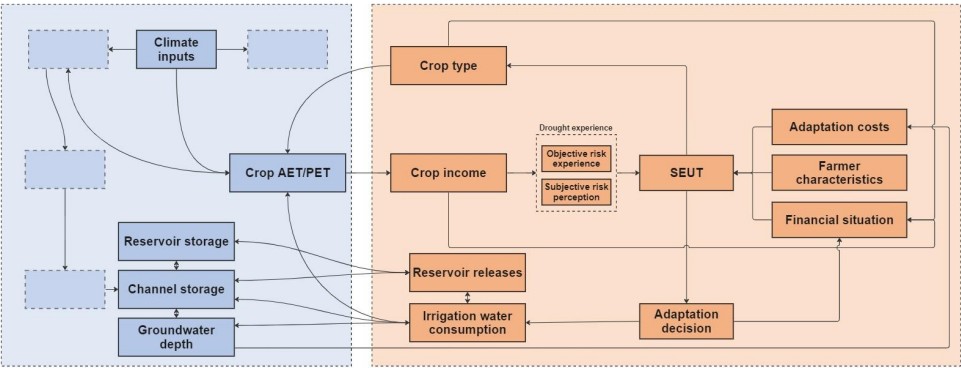

**Figure 1 Simplified model setup integrating the hydrological model CWatM (blue boxes) with an agent-based model (orange boxes).**


**2.1 Model setup.**
Figure 1 shows the structure of the GEB model. In short, GEB couples a large-scale agent-based model
(orange part) that simulates the adaptation behavior of millions of agents (farmers and reservoir operators) (De
Bruijn et al., 2023) to a hydrological model (blue part) simulated with the CWatM (Burek et al., 2020) and



MODFLOW models (Langevin et al., 2017). The hydrological processes of CWatM operate at daily timesteps at
30 arcsec grid size, while GEB's agent processes are at sub-grid level. The interaction between both, such as
irrigation, occurs daily, while adaptation decisions are made at the end of each growing season for the next one.
The CHELSA-W5E5 v1.0 observational climate input data at 30 arcsec horizontal and daily temporal resolution
was used as climate forcing (Karger et al., 2022). The agent's individual characteristics are derived from socio-
economic data (census data on e.g. income), survey data (on e.g. risk aversion, discount rate), agricultural data
(past yields, crop rotations, farm sizes) and data on past climate and droughts (SPEI) (section 2.3-2.5 and B.1 to
B.4). These data are used to calculate the Subjective Expected Utility (SEUT) equation to determine whether a
farmer adapts or not, given the hydro-climatic context.
**2.2 Case study.**
The Upper Bhima catchment in Maharashtra, spanning 45,678 km², varies in elevation from 414 m in the east to
1458 m in the Western Ghats mountain range (Figure 2). The catchment is mostly flat, with 95 % of its area below
800 m. The area experiences significant rainfall variation due to interaction of the monsoon and the Western Ghats,
ranging from 5000 mm in the mountains to less than 500 mm in the east (Gunnell, 1997). Over 90 % of this rain
falls during the monsoon months (June–September), with substantial deficits from October to May. The state's
agricultural cycle includes the monsoon Kharif season (June–September) and the dry Rabi season (October–
March), with April and May constituting the hot summer period.

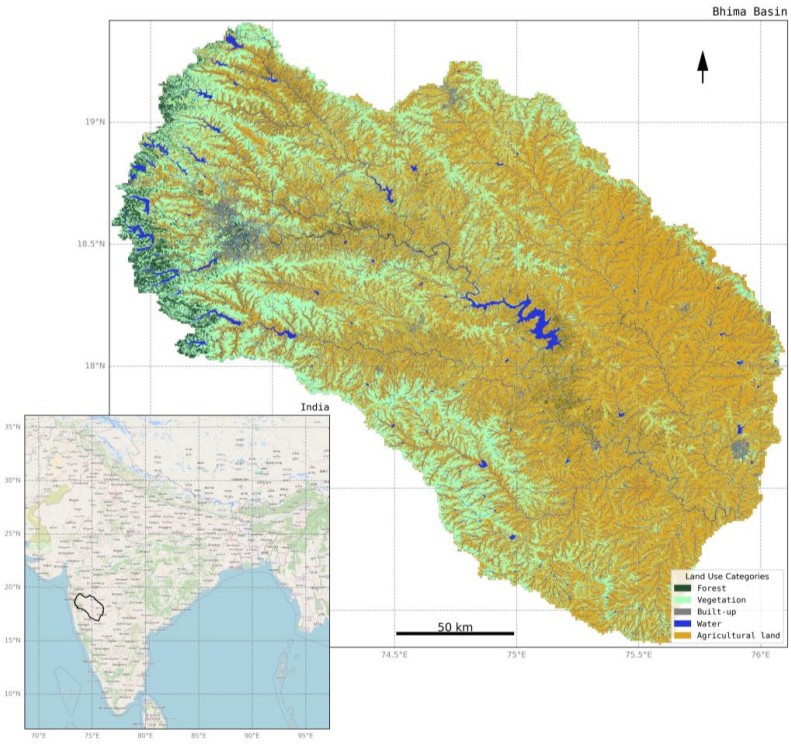




**Figure 2 Overview of the Bhima basin's location in India and the land use classification used in the model. The forested**
**area in the west are the Western Ghats mountain range. Map of the Bhima basin land cover produced from land-cover**
**data from Jun et al. (2014). © OpenStreetMap contributors 2024. Distributed under the Open Data Commons Open**
**Database License (ODbL) v1.0.**
To manage water supply, reservoirs in the Western Ghats accumulate water during monsoon rains. This water is
released to the river and to farmers in the reservoir command areas through a system of canals during the monsoon
(Kharif) and the dry irrigation season (Rabi & Summer). This results in human-controlled river flows, which are
less dependent on natural climate patterns (Immerzeel et al., 2008). Although reservoirs distribute irrigation water,
agriculture in Maharashtra still mainly relies on monsoon rain, with 19.7% of the state's gross cropped area being
irrigated and 80.2 % dependent on rainfed farming (Udmale et al., 2015). During the study period there were
approximately three periods with a prolonged negative 12-month Standardized Precipitation Evapotranspiration
Index (SPEI) score: a severe- (2000-2005), mild- (mid-2009 to 2010), and a last moderate-mild (mid-2012 to 2015)
drought (McKee et al., 1993). The middle of the last drought experienced a brief period of positive SPEI, but for
ease of referencing we refer to it as one drought.

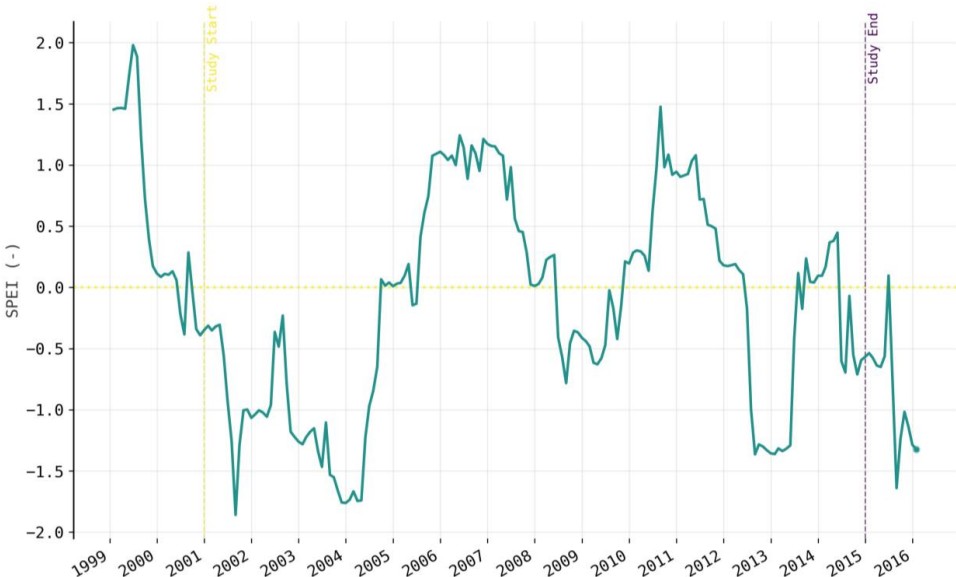


**Figure 3 The average 12-month Standardized Precipitation Evaporation Index (SPEI) in the Bhima basin. Derived**
**from the CHELSA-W5E5 v1.0 dataset (Karger et al., 2022).**


**2.3 Farmer decision rules**
Agents make decisions based on the SEUT (Fishburn, 1981), which has been widely used in various ABMs to
simulate adaptive behavior. (Groeneveld et al., 2017; Haer et al., 2020; Tierolf et al., 2023; M. Wens et al., 2020).
A major advantage of the SEUT is that it facilitates economic maximization while accounting for an individual's
subjective characteristics (i.e. risk aversion and discount rate) and dynamic risk perception that adjusts in response
to drought events. At each yearly timestep agents calculate the following (S)EUTs:






1.    SEUT of taking no action (Eq. 1)
2.    SEUT of investing in a (tube-) well (Eq. 2)
3.    SEUT of their current crop rotation (Eq. 3)
4.    EUT of their current crop rotation (Eq. 4)

To decide whether to invest in a well, agents compare the SEUT of taking no action with the SEUT of digging a
well. When the SEUT favors adaptation and adapting is within the agent's budget constraints, the farmers invest
in a well. With respect to crop rotation, there are over 300 unique crop rotations used within the model. It would
be computationally unfeasible for each agent to calculate the SEUT for each rotation. Furthermore, literature shows
that people tend to emulate their neighbors' practices (Baddeley, 2010; Haer et al., 2016). Therefore, all agents
calculate only their own crop rotation's SEUT (Eq. 3) and EUT (Eq. 4, using neutral risk perception, aversion and
discount rate, section 2.5). Then, agents compare their current crop rotation's SEUT with the EUT of all
neighboring farmers using similar irrigation sources (within a 5 km radius, using reservoir, surface, groundwater
or no irrigation). The EUT is used since using a neighbor's SEUT would mean using another agent's subjective
factors. They then adopt the crop rotation of the neighbor who's EUT is highest, if this exceeds their own SEUT.
$$SEUT_{no\_action} = \int_{p_2}^{p_1} \beta_t * p_i * U \left( \sum_{t=0}^{T} \frac{Inc_{i,x,t}}{(1+r)^t} \right) dp \qquad (1)$$
$$SEUT_{tube\_well} = \int_{p_2}^{p_1} \beta_t * p_i * U \left( \sum_{t=0}^{T} \frac{Inc_{i,x,t}^{adapt}-C_{t,d}^{adapt}}{(1+r)^t} \right) dp \qquad (2)$$
$$SEUT_{own\_crop\_rotation} = \int_{p_2}^{p_1} \beta_t * p_i * U \left( \sum_{t=0}^{T} \frac{Inc_{i,x,t}-C_{t,m}^{input}}{(1+r)^t} \right) dp \qquad (3)$$
$$EUT_{own\_crop\_rotation} = \int_{p_2}^{p_1} p_i * U \left( \sum_{t=0}^{T} \frac{Inc_{i,x,t}-C_{t,m}^{input}}{(1+r)^t} \right) dp \qquad (4)$$

Utility *U(x)* is a function of expected income *Inc* and potential adapted income *Inc^adapt* per event *i* and adaptation
costs *C^adapt*. In eq. 2, *C^adapt* is dependent on groundwater levels and in eq. 4 on current market prices. To calculate
the utility of all decisions, we take the integral of the summed and time (*t*, years) discounted (*r*) utility under all
possible events *i* with a probability of $p_i$ and adjust $p_i$ with the subjective risk perception $\beta_t$. See table B1 for an
overview of all model parameters.
*Predicted income:* To calculate the expected utility, we need information on farmer income during
droughts of varying return periods with and without an adaptation. Since droughts of similar return periods have
different severities depending on the farmer's location, and since this relation is also dependent on each farmer's
crop rotation and irrigation capabilities, no straightforward empirical relationship exists. Therefore, we established
this relationship endogenously for each farmer in the following manner. After each harvest, the 12-month SPEI
(derived from the CHELSA climate data between 1979 and 2016) at the time of harvest and the harvest's yield
ratio (section 2.4) are determined for each agent. The SPEI is converted to a drought probability and these values
are then averaged per year. In order to get more data points, they are then averaged per farmer group, which are
based on farmers' elevation (upstream, midstream, downstream), irrigation (well or no well) and crop rotation.
Then, a relation (eq. 5) is fitted between drought probability and yield ratio for each group using the last 20 years



of data (a spin-up period of 20 years is used where no behavior occurs). We refer to this relation as the agent's
objective drought risk experience. The 12-month SPEI and base 2 logarithm were chosen as they returned the
highest R-squared between drought probability and yield ratio for this region (~ 0.50).

$$SPEI_{i,t} = a * log_2(yield_{i,t}) + b \qquad (5)$$


The relation between probability and yield ratio is used to derive yield ratios associated with 1, 2, 5, 10, 25 and
50-year return period drought events $i$, which are then converted to income per return period event $Inc_i$ (section
2.4). To determine their potential income after adaptation $Inc^{adapt}$, within groups of similar cropping and elevation,
the non-irrigating groups determine their yield ratio gain from the yield ratios of their well-irrigating counterparts.

*Cost of wells:* To determine the cost of wells, we adapted the cost equations and parameterization of

Robert et al. (2018) (Appendix B.1). These are a function of pump horse power, pumping hours, electricity costs,
probability of well failure, maintenance costs and drilling costs. Drilling costs are dynamic and dependent on the
well's depth, which are put at 20 m below the current groundwater table. Together with the agent's interest rate $r$
(section 2.4, B.2), this is converted to an annual implementation cost $C^{adapt}$ for the n-year loan using eq. 6.

$$C_{t,d}^{adapt} = C_d^{fixed\ cost} * \frac{r*(1+r)^n}{(1+r)^n-1} + C_t^{Yearly\ costs} \qquad (6)$$


*Crop costs:* Yearly cultivation input costs $C^{input}$ per hectare for each crop type, which include expenses

such as purchasing seeds, manure, and labor are sourced from the Ministry of Agriculture and Farmers Welfare.
(https://eands.dacnet. Nic.in/Cost_of_Cultivation.htm, last access: 15 July 2022) (De Bruijn et al., 2023).

*Loans and budget constraints:* We assume that agents are "saving-down" (Bauer et al., 2012) and taking

loans for agricultural inputs (Hoda & Terway, 2015) and investments using eq. 6. We assume farmers cannot spend
their full income on inputs and investments and implement an expenditure cap (Hudson, 2018), which we use as a
calibration factor (section 2.6). If the proposed annual loan payment for a well exceeds the expenditure cap, agents
are unable to adapt. Chand et al. (2015) put expenditure of inputs such as seeds, fertilizer, plant protection, repair
and maintenance feed and other inputs at approximately 20-25%. Thus, including the extra well investments cost,
we calibrate the expenditure cap of yearly payments between 20-50% of yearly non-drought income (Pandey et
al., 2024).

*Time discounting and risk aversion:* For eq. 1-3 the agent's individual discount rate and risk aversion

(section 2.5) are used. For eq. 4, as the goal is a "neutral" expected utility of farmer's crops, all farmers use the
average discount rate and risk aversion. For eq. 1-2 a time horizon of 30 years following Robert et al. (2018) is
used, while for eq. 3-4 a time horizon of 3 years is used. The utility $U(x)$ as a function of risk aversion $\sigma$ is as
follows:

$$U(x) = \frac{x^{1-\sigma}}{1-\sigma} \qquad (7)$$


*Bounded rationality:* Bounded rationality is described by the risk perception factor $\beta$. $\beta$ rises after agents

have experienced a drought, overestimating drought risk ($\beta > 1$). After time without a drought, it lowers again,




underestimating risk (*β < 1*). We follow the setup of Haer et al. (2020 and Tierolf et al. (2023) and define *β* as a
function of *t* years after a drought event:

$$\beta_t = c * 1.6^{-d*t} + e$$                        (8)

We set *d* at -2.5, resulting in a slower risk reduction than in Haer et al. (2020) and Tierolf et al. (2023), as farmers
are assumed to retain more awareness of drought risk compared to households of flood risk (van Duinen et al.,
2015). We set the minimum underestimation of risk *e* at 0.01 and calibrate the maximum overestimation of risk *c*
between 2 and 10 (Botzen & van den Bergh, 2009).

*Drought loss threshold:* As the onset of droughts are not as obvious as with floods (van Loon et al., 2016),
we define an agent's drought event perception (Bubeck et al., 2012) according to a loss in yield ratio against a
moving reference point, similar to prospect theory (Kahneman & Tversky, 2013; Neto et al., 2023). The moving
reference point is the 5-year average difference between the reference potential yield and the actual yield (2.4).
We calibrate the drought loss threshold between 5% and 25%. This means that if the current harvest's difference
between potential and actual yield falls 5-25% below the historical average, the years since last drought event *t*
(Eq. 8) is reset and *β* rises.

*Microcredit:* If the yield falls below the drought loss threshold, agents will also take out a loan equal to the
missed income (P. D. Udmale et al., 2015). The loan duration is set at 2 years (Rosenberg et al., 2013).

### 2.4 Farmer crop cultivation

*Yield & Income:* Farmers grow pearl millet, groundnut, sorghum, paddy rice, sugar cane, wheat, cotton,
chickpea, maize, green gram, finger millet, sunflower and red gram. Each crop undergoes four growth stages (d1
to d4). The crop coefficient (Kc) is then calculated as follows (Fischer et al., 2021):
$$Kc_t = \begin{cases} Kc1, & t < d_1 \\ Kc1 + (t - d1) \times \frac{Kc2-Kc1}{d2}, & d_1 \le t < d_2 \\ Kc2, & d_2 \le t < d_3 \\ Kc2 + (t - (d1+d2+d3)) \times \frac{Kc3-Kc2}{d4}, & \text{otherwise;} \end{cases}$$                  (9)

where *t* represents the number of days since planting, and d1 to d4 are the durations of each growth stage. At the
harvest stage, the actual yield (Ya) is determined based on a maximum reference yield (Yr; Siebert & Döll, 2010),
the water-stress reduction factor (KyT), and the ratio of actual evapotranspiration (AET) to potential
evapotranspiration (PET) throughout the growth period (Fischer et al., 2021):
$$Y_a = Y_r \times \left(1 - KyT \times \left(1 - \frac{\sum_{t=0}^{t=h} AET_t}{\sum_{t=0}^{t=h} PET_t}\right)\right)$$               (10)

We refer to the latter part of Eq. 10 as the "yield ratio", i.e., the fraction of maximum yield for a specific crop.
Actual yield is then converted into income based on the state-wide market price for that particular month. Historical



monthly market prices are sourced from Agmarknet (https://agmarknet.gov.in, last accessed on 27 July 2022) (De
Bruijn et al., 2023).
*Irrigation:* The irrigation demand for farmers is calculated based on the difference between the field
capacity and the soil moisture, and it is restricted by the soil's infiltration capacity (De Bruijn et al., 2023). If
agents have access to all irrigation sources, they first meet their demand using surface water, followed by
reservoirs, and finally groundwater. When a farmer opts to irrigate, the necessary water is drawn from the
appropriate sources in CwatM and subsequently dispersed across the farmer's land.

### 2.5 Agent initialization

*Agent initialization:* To generate heterogeneous farmer plots and agents with characteristics statistically
similar to those observed within the Bhima basin, factors from the IHDS (Desai et al., 2008), such as agricultural
net income, farm size, irrigation type or household size, were combined with Agricultural census data (Department
of Agriculture & Farmers Welfare India, n.d.). For this, we use the iterative proportional fitting algorithm, which
reweights IHDS survey data such that it fits the distribution of crop types, farm sizes and irrigation status at sub-
district level reported in the Agricultural Census (De Bruijn et al., 2023). The farmer agents and their plots were
randomly distributed over their respective sub-districts on land designated as agricultural land (Figure 3; Jun et
al., 2014) at 1.5″ resolution (50 meter at the equator).
*Risk aversion & discount rate:* To set risk aversion and discount rate, we first normalized the distribution
of agricultural net income. Then, as risk aversion and discount rate correlate with household income (Bauer et al.,
2012; Just & Lybbert, 2009; Maertens et al., 2014), we rescaled the normalized income distribution with the mean
and standard deviation of the (marginal) risk aversion $\sigma$ (0.02, 0.82; Just & Lybbert, 2009) and discount rate *r*
(0.159, 0.193; Bauer et al.2012) of Indian farmers. Noise was added to both to prevent that each present-biased
agent is also risk taking by definition.
*Interest rates:* To account for the variation in access to credit and interest rates among farmers, we
assigned each agent an interest rate based on their total landholding size, with smaller farmers receiving higher
and larger farmers lower rates (Appendix B.2, Maertens et al., 2014; P. D. Udmale et al., 2015). This assignment
is based on the interest rates observed among Indian farmers (Hoda & Terway, 2015; P. D. Udmale et al., 2015).

### 2.6 Calibration, validation, sensitivity analysis and runs

*Calibration:* We calibrated the model from 2001 to 2010 using observed daily discharge data and yield
data. The full data range of available observed data was used to calibrate the model, following the
recommendations of Shen et al. (2022), which found that
calibrating fully to historical data without conducting model validation is the most robust approach for hydrological
models. The daily discharge data was obtained from 5 discharge stations at various locations in the Bhima Basin.
The yield data was obtained by dividing the total production by the total cropped area from ICRISAT (2015) to
determine yield in tons per hectare. This figure was then divided by the reference maximum yield in tons per
hectare to calculate the percentage of maximum yield, aligning with the latter part of Eq. 10. Calibration is done
for several standard hydrological parameters, including the maximum daily water release from a reservoir for
irrigation, typical reservoir outflow, and the irrigation return fraction (Burek et al., 2020). Furthermore, it was done
for the expenditure cap, base yield ratio, drought loss threshold and the maximum risk perception (Appendix B.3).



The process utilizes the NSGA-II genetic algorithm (Deb et al., 2002) as implemented in DEAP (Fortin et al.,
2012), to optimize the calibration based on a modified version of the Kling-Gupta efficiency score (KGE; Eq. 11;
Kling et al., 2012), similar to (Burek et al., 2020, De Bruijn et al., 2023).

$$\mathrm{KGE}' = 1 - \sqrt{(r-1)^2 + (\beta - 1)^2 + (\gamma - 1)^2} \qquad (11)$$

Where $r$ is the correlation coefficient between monthly and daily simulated and observed yield ratio and discharge,
respectively. $\beta = \frac{\mu_s}{\mu_0}$ represents the bias ratio, and $\gamma = \frac{CV_s}{CV_0} = \frac{\sigma_s \mu_s}{\sigma_0 \mu_0}$ is the variability rate. The optimal values for $r$,
$\beta$ and $\gamma$ are 1. The final KGE scores were $\pm$ 0.63 for the discharge and $\pm$ 0.60 for the yield.
*Sensitivity analysis:* A Delta Moment-Independent Analysis with 300 distinct samples was done using
the SALib Delta Module (Iwanaga et al., 2022). Risk aversion, discount rate, interest rate, well cost, and the
drought loss threshold were varied to assess their impact on well uptake, crop income, yield, risk perception,
groundwater depth, reservoir storage, and discharge upstream and downstream. For detailed parameter settings,
refer to Appendix B.4.
*Model runs & scenarios:* The model had a spin-up period from 1980 to 2001, and ran from 2001 to 2015.
The periods with a prolonged negative 12-month SPEI during this period were: a severe- (2000-2005), mild- (mid-
2009 to 2010), and a moderate-mild (mid-2012 to 2015) drought (McKee et al., 1993). Two scenarios were run:
one without adaptation, where agents maintained the same crop rotation and irrigation status as at the start of the
model, and another where agents could change their crops or dig wells according to the decision rules outlined in
section 2.3. To account for stochasticity, both scenarios were run 60 times, after which the average results and the
standard error of the mean were calculated.




**3 Results**
**3.1 Crop switching and well uptake in the Adaptation scenario**

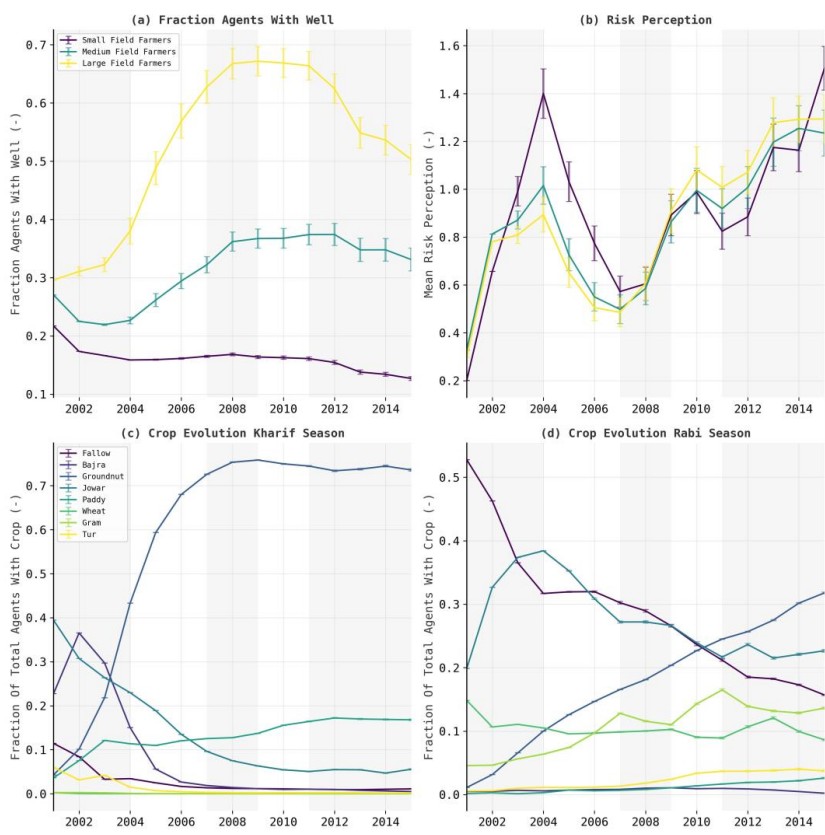


**Figure 4 Evolution of Wells, Risk Perception and Crops in the Bhima basin. (a-b) Farmers are categorized by field size into small (0-33rd percentile), medium (33-67th percentile), and large (67-100th percentile) groups; (a) the fraction of the total group with a well; (b) the mean Risk Perception of each group. (c-d) Evolution of the dominant crops in the wet Kharif (c) and dry Rabi (d) season. Values are 60 run means (a-d), error bars indicate standard error  (a-b), light grey areas indicate years where the average 1 month Standardized Precipitation Evaporation Index (SPEI) was below 0.**


Figure 4 shows how agent characteristics change over time for three different field sizes: large scale (67-100
percentile of size, >1.8 ha; yellow), medium scale (33-67 percentile of size, 0.82-1.9 ha; blue), and small scale (0-
33 percentile of size, <0.82 ha; purple) farmers. Panel 4a shows that for large scale farmers adaptation first slowly
rises and speeds up after the first drought (2001-2004), alongside an increase in risk perception from the first
drought. For medium farmers, well uptake initially decreases but then increases alongside a similarly heightened
risk perception. For smallholder farmers, the number of well owners declines and then only slightly recovers after
the first drought, even though they have a higher risk perception compared to medium and large field farmers. This
difference between well owners mirrors the differences in interest rates, where smallholder farmers have the
highest interest rates on loans, and large farmers the lowest rates (Appendix A.1). This highlights that loan interest



is an important factor in whether agents adapt. During the last drought (2011-2015), despite high-risk perception,
the proportion of farmers owning wells declines across all farm sizes (figure 4a-b). The adaptation by large farmers
declines the steepest, although they do remain the most adapted group (Section 3.2).

In the Kharif wet season, all crop types except paddy-irrigated rice and groundnut decrease in prevalence (Figure
4c). Both groundnut and paddy cultivation have steeply risen in profitability during the study period (7g), however,
paddy cultivation is substantially more water intensive than groundnut. In the dry Rabi season we see a large
decrease of farmers who leave their field fallow (i.e. no crops), which is mainly replaced by cultivating groundnut,
although there is a greater heterogeneity of cultivated crops in the Rabi season as compared to the wet Kharif
season (Figure 4d). Furthermore, the increase and decrease of Jowar cultivation, which is less water-intensive
compared to Groundnut and Paddy irrigation and performs well during droughts (Singh et al., 2011), aligns very
well with drought and non-drought periods. Lastly, we see almost no Paddy cultivation in the dry season.

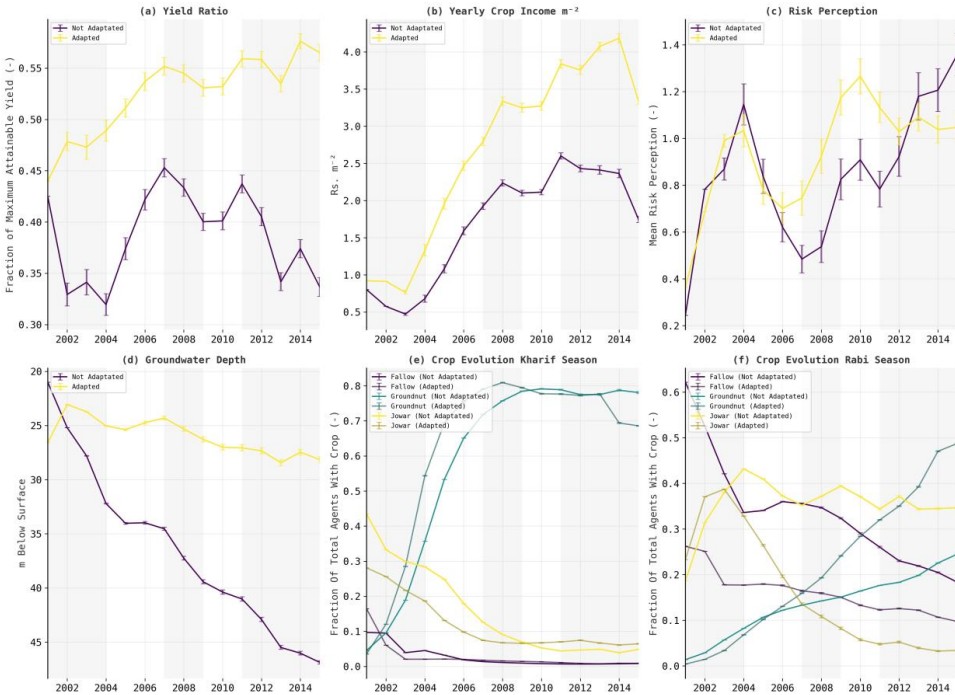


**Figure 5 Evolution of Yield ratio, Income, Risk perception, Groundwater Depth and the two main crops in the Wet Kharif and Dry Rabi Season in the Bhima basin. (a-d) Farmers are categorized by whether they have wells in each year into a Not Adapted and Adapted group. Light grey areas indicate years where the average 1 month Standardized Precipitation Evaporation Index (SPEI)was below 0.**


Figure 5a shows a large difference in yield ratio between farmers with- or without a well, likely stemming from
the increased water reliability due to irrigation wells. Consequently, farmers with wells saw a yield ratio increase
instead of decrease during the first drought. Yearly crop income is approximately 30 % higher for farmers with
wells (5b), though incomes for both groups have increased due to switching to higher-priced crops. Importantly,
this data does not only show the effects of wells, but also which farmers are able to initially afford wells, stemming





from prior higher yield, income and lower groundwater levels. Groundwater levels are unexpectedly higher for
farmers with wells (5d), despite wells being the primary cause of groundwater depletion for most farmers (6d, 7c).
However, note that in the figure, farmers whose well dried up count as Not Adapted. Thus, when farmers with
wells are in locations where groundwater recharge cannot keep up with extraction, their wells dry and they are
switched to the Not Adapted group. Subsequently, only farmers with wells where groundwater is not rapidly
depleted, or those who have recently installed wells, remain in the Adapted group, resulting in high average
groundwater levels for this group. The extraction and hydroclimatic conditions at the farmers' locations where
depletion matches the Adapted group's average thus provide an estimate of the necessary circumstances to
sustainably maintain wells. As long as these conditions are present, the increased yield ratios and income (5a-b)
can be maintained.

Figures 5e and 5f depict the development of Fallow, Jowar, and Groundnut cultivation during the wet Kharif and
dry Rabi seasons. We show these crops as they are most widely cultivated and dynamic (Figure 4). In the Kharif
season, crop patterns are similar for both groups (5e). During the Rabi season, both agents with and without wells
switch to Jowar during the first drought (2001-2004, 5f). However, after the initial drought, the percentage of
agents with wells cultivating Jowar massively reduces, while the fraction without wells cultivating Jowar remains
stable. Furthermore, during Rabi, more adapted agents cultivate Groundnut, while fewer leave their land fallow.
This contrast in cultivation patterns among well-irrigating and non-irrigating groups highlights the critical role of
water availability in agent's crop selection. If rainfall is ample, such as during the wet season, the patterns between
farmers with and without wells are similar. However, in drier conditions, these patterns diverge because farmers
with wells have greater water availability. This aligns with the patterns seen in Figure 4.





**3.2 Crop switching and well uptake in the Adaptation vs. the No Adaptation scenario**

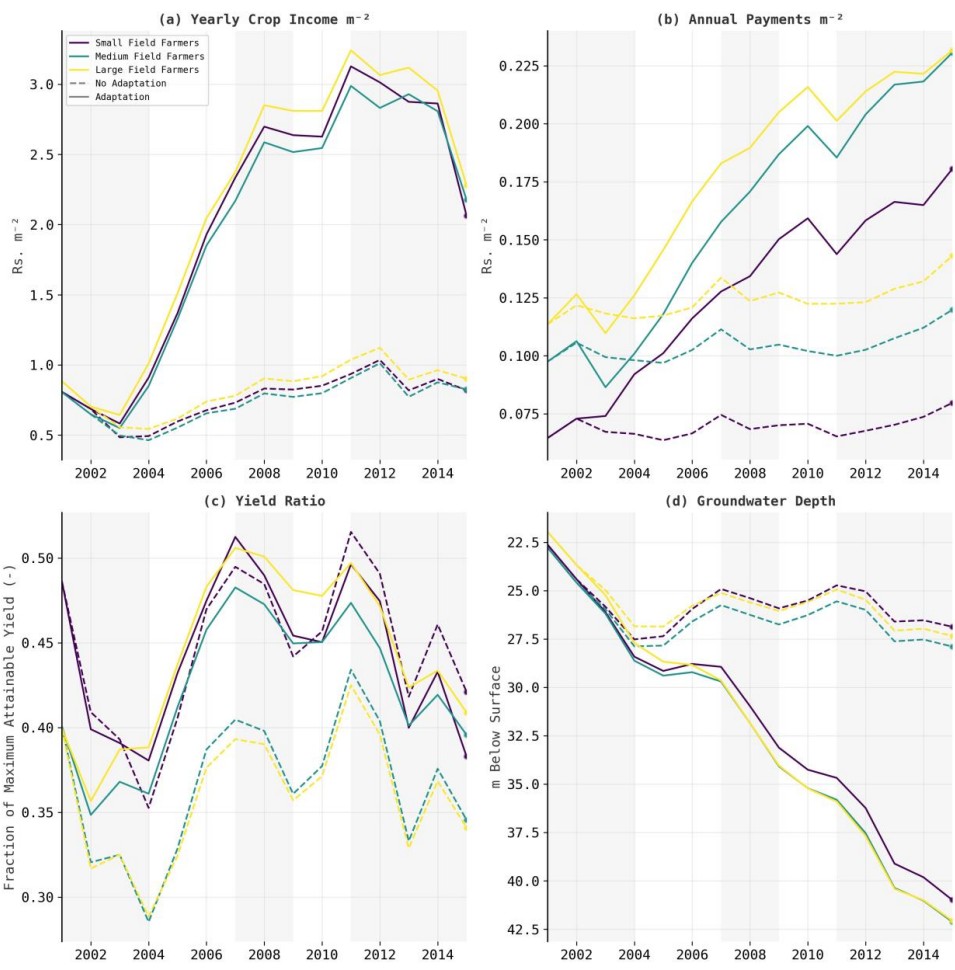


**Figure 6 Evolution of Income, Loan Payments, Groundwater Depth and Yield Ratio in the Bhima basin for a scenario where agents adapt (filled line) and where they stick to their initial adaptations and crops (dotted lines). (a-c) Farmers are categorized by field size into small (0-33rd percentile), medium (33-67th percentile), and large (67-100th percentile) groups; (a) Inflation adjusted early Income in Rs / m2 after harvesting and selling crops; (b) Inflation Adjusted Yearly Loan Payments in Rs / m2, consisting of payments for cultivation costs, well loans and microcredit in case of crop failure; (c) Average yield ratio of agent groups; (d) Groundwater Depth in m below surface. Values are 60 run means (a-d), light grey areas indicate years where the average 1 month Standardized Precipitation Evaporation Index (SPEI) was below 0.**


Figure 6 compares a scenario where agents adapt (i.e., switch crops or dig wells) to one where agents stick with
their initial adaptation. Figure 6a shows that despite the increased well uptake for larger farmers, the average
income per square meter varies by no more than 5 % between farm size groups, which contrasts the difference
shown in Figure 5-b. This is illustrated by the yield ratio (6c), where initially, smaller farmers achieve substantially
higher yields than larger farmers due to cultivating crops with lower water demand. Once larger farmers switch
crops and install more wells, yields match or exceed those of smaller farmers.




During the first and most severe droughts from 2001 to 2004, the drop in yield ratio of the no-adaptation scenario
was six times worse (5 % versus 30 % drop, figure 6c). These initial yield gains were likely due to a shift towards
less water-intensive crops (Jowar), as for medium field size farmers yields also increased, while their well uptake
declined (Figure 4a, 6c). Subsequent yield increases align better with well uptake, with larger farmers achieving
higher yields than smaller ones. Furthermore, after the initial drought period, larger farmers switched to higher
grossing but more water intensive crops (4d), as the yield ratios between small and large farmers were similar,
while profits were higher. However, ultimately, well uptake dropped (Figure 4a). Consequently, during the last
drought from 2011 to 2015, the relative yield drop for larger farmers was similar across both the adaptation and
no-adaptation scenarios, contrasting with the six times decrease seen during the first drought. Furthermore, the
income fell 10-20 % more in the adaptation scenario.

For larger farmers with access to low interest loans (Appendix A.1), the annual cost to invest in wells is a smaller
percentage of the agents' income. The influence of this 'effective investment cost m$^{-2}$' (Sayre & Taraz, 2019) is
reflected in the annual loan payments m$^{-2}$ in Figure 4b, where the payments are equal for the medium and large
farmers, while the large farmers have a higher fraction of adapted agents (Figure 4a). Moreover, even compared
to smaller farmers—who have 80-84% fewer adapted agents—the annual payments m$^{-2}$ are not substantially
higher. Lastly, the annual payments m$^{-2}$ are lower than what the expenditure cap ($\pm$ 29 % of income) would suggest
(Figure 4b). This likely results from using group averages, where not adapted agents with smaller loans lower the
average, and from using non-drought income based on the yield-probability relation instead of the most recent
incomes. The latter adjusts more slowly to increased income, making agents more risk averse. Switching to using
the most recent incomes could change this.

In Figure 6d, the groundwater levels in the no-adaptation scenario drop 5 meters between 2001-2004 and then
stabilizes. Conversely, in the adaptation scenario, groundwater levels continue to decrease by an average of 1 meter
annually, stabilizing briefly during periods of positive SPEI (i.e., no droughts) and declining rapidly during
droughts. The rate of groundwater decline is roughly the same for all farmers, regardless of farm size. The most
recent rapid decline in 2011 corresponds with a decrease in well uptake (Figure 4a), suggesting that this decline is
primarily due to wells drying up. Since larger farmers were the early adopters, their shallower wells were the first
to dry up, which explains their more rapid decline compared to medium and small farmers (Figure 4a). However,
despite declining well uptake, loan payments remain high due to ongoing loans.





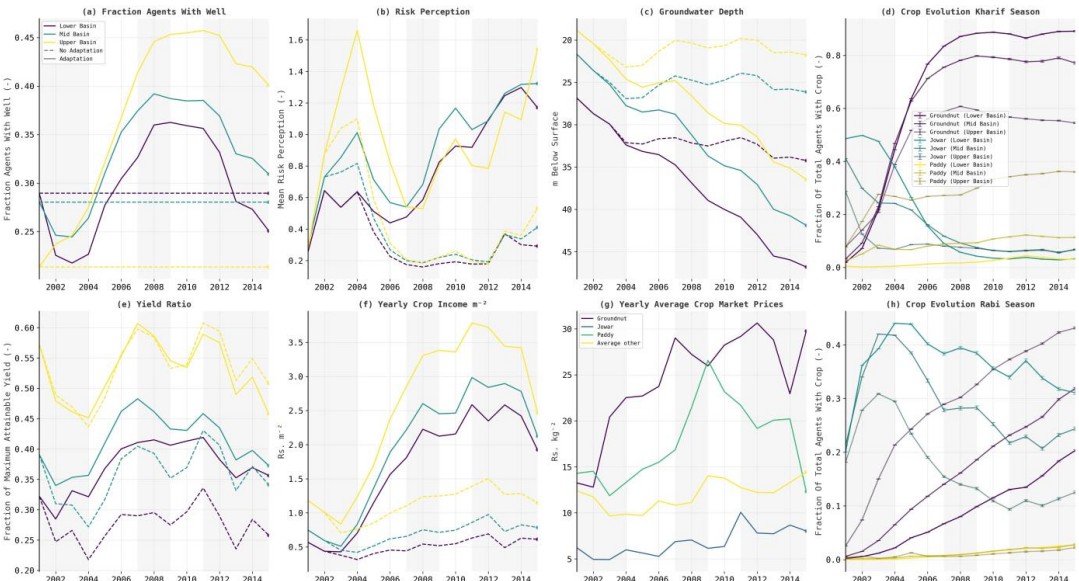


**Figure 7 Evolution of Wells, Risk Perception, Groundwater Depth, the two most cultivated crops in the Wet Kharif and Dry Rabi season, Yield and inflation adjusted Yearly Crop Income and Observed Crop Market Prices in the Bhima basin. Farmers are categorized by farmer elevation into Lower Basin (0-33rd percentile), Mid Basin (33-67th percentile), and Upper Basin (67-100th percentile) groups (a-c, e-f). Values are 60 run means, light grey areas indicate years where the average 1 month Standardized Precipitation Evaporation Index (SPEI) was below 0.**


In Figure 7, farmers are categorized as upstream (67-100th percentile elevation), midstream (33-67th percentile),
and downstream (0-33th percentile). Mid- to downstream farmers initially see a reduction in well use, with
increases only occurring at the end of the first drought (2001-2004, Figure 7a). This aligns with increased incomes
late in the first drought as a result of the drought ending and switching to more profitable crops (7g). The crop
switching has a dual effect: firstly, it boosts income, enabling agents to invest more in wells; secondly, it enhances
well profitability, as now the same amount of water leads to a larger absolute increase in income. Upstream, the
initial yield, income and groundwater levels are higher. Higher groundwater levels reduce the price of wells and
higher incomes increase what agents can spend on wells. Similar to what was seen for larger farmers in Figures 4
and 6, this reduces the effective investment costs, meaning the wells cost a smaller percentage of the agents'
income and more agents adapt. This causes upstream farmers to immediately adapt as the model starts, even during
the first drought (2001-2004). Similar to the trends in Figure 6d, groundwater levels quickly drop during droughts
and stabilizes when SPEI is positive. This pattern is mirrored in well uptake, which increases until 2007 but halts
in 2008, coinciding with a sharp decline in groundwater during the middle drought (2007-2009). During the last
drought (2011-2015), groundwater levels rapidly fall again and well uptake substantially declines due to wells
drying up. This decline intensifies downstream, resulting in downstream farmers having fewer wells than they
initially had.

Despite fewer wells among downstream farmers, groundwater levels decline similarly to those in the mid and
lower basins (Figure 7c). Comparing this against spatially varying parameters between the lower-, mid- and upper
basin, we mainly see that upstream agent density is lower and precipitation is higher (Appendix A.2). In the upper





basin this means less additional irrigation water is required, resulting in more recharge and less agents abstracting
groundwater per km². This also correlates with the shown higher yield and income (Figures 7e-f).

During the wet Kharif season, mid- and downstream farmers grow almost solely groundnut, whereas upstream
paddy cultivation is also common (Figure 7d). This follows the earlier shown pattern of higher water availability
generally leading to more water intensive crops. The yield ratio is highest upstream and lowest downstream, with
downstream also showing a greater difference in yield between the adaptation and no-adaptation scenario (Figure
7e). This may be the effect of higher water demand upstream, which is caused by more water-intensive crops
offsetting more of the supply gains. This is also reflected in a lower yield ratio compared to the no-adaptation
scenario, even though there are more agents with wells.

For mid- and downstream farmers, yield ratios increased during the first drought compared to the no-adaptation
scenario, even though well uptake declined (Figure 7a, e). Similar to what was discussed at Figures 4-6, this
increase was due to a shift toward a less water-intensive crop (Jowar, 7h). Subsequently, as water availability
increased, the prevalence of Jowar declined, while Groundnut, which requires more water than Jowar but less than
Paddy, continued to rise due to its steep price increase (7g). This pattern again followed water availability, as this
was more pronounced for the mid- and upstream farmers. The economic maximalization through crop switching
boosted incomes without requiring additional water from wells (7a, 7f). However, yields in the adaptation scenario
for mid- and downstream farmers continued to rise compared to the no-adaptation scenario. Furthermore, both
yields fell less during the middle drought. This pattern aligns with the initial rise well usage for these groups (7a).
Ultimately, well uptake fell, and during the last droughts (2011-2015) yield ratios fell by 18-22 %, approximately
equally as much as in the no-adaptation scenario. However, from 2011 to 2015, crop income in the adaptation
scenario fell by 25-35%, a 10-15% greater decline compared to the no-adaptation scenario. This is a larger fall
than what only the yield ratios would suggest, and can be explained by a simultaneous drop in prices for the main
cultivated crops (7g).



**3.3 Sensitivity Analysis**

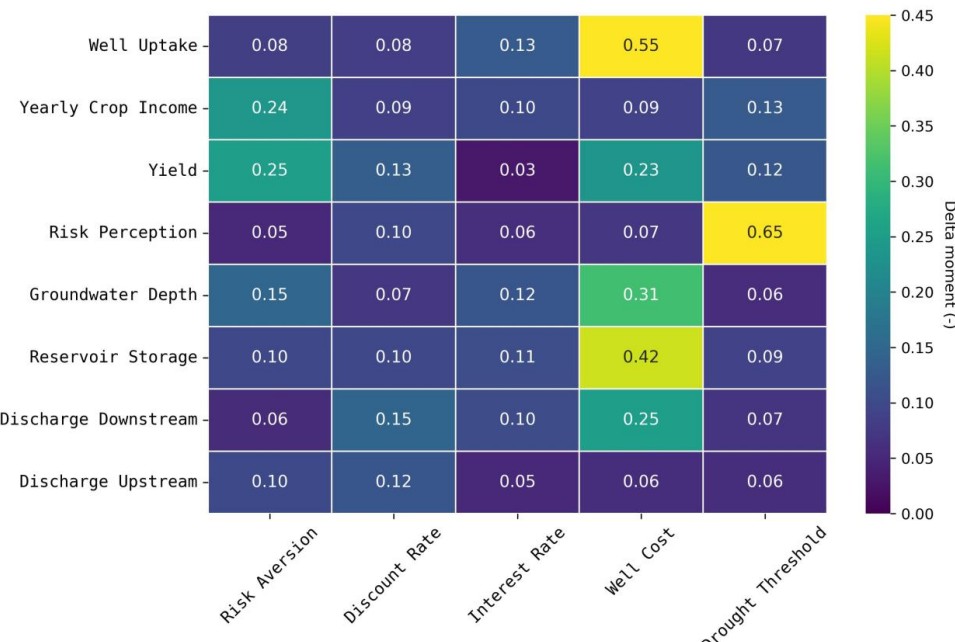

**Figure 8 Delta moment Sensitivity Analysis. Values indicate how sensitive an output factor (y-axis) is to the influence of a specific input factor (x-axis), in relation to the influence of all other input factors. The output consists of number of wells, yearly crop income, yield, risk perception, groundwater depth, reservoir storage and discharge up- and downstream. The changed input parameters consist of risk aversion, discount rate, interest rate, well cost and drought threshold.**


Our results show that well uptake is highly sensitive to well cost. Diving deeper in this relation, Figure 8 shows
that although well cost substantially affects the adoption of wells and yield, its impact on income is minimal
compared to other factors. This notion is supported by Figures 4 to 7 who reveal that many farmers cannot afford
wells regardless of cost changes and that decreasing groundwater levels result in the loss of wells for more. Thus,
although the effect of wells is large for farmers with wells (Figure 4), there remains a large group without wells
throughout the basin. In contrast, risk aversion substantially affects both well adoption and crop selection, and
crop selection is relevant for all farmers. Furthermore, crop selection is especially impactful as the price of
groundnut, the primary crop farmers switch to in the main season, doubled relative to other crops (Figure 7g). This
illustrates that farmer's  adaptive behavior is a mix of climate and market dynamics.

However, Figure 8 shows that well cost substantially influences all hydrological parameters except upstream
discharge. Recorded in regions with higher precipitation and fewer agents (Appendix A.2), upstream discharge
shows little sensitivity to well cost, suggesting groundwater extraction makes up a smaller fraction of total river
inflow. Similar to income, yield reacts to risk aversion through crop choice. Risk perception is sensitive to the
drought loss threshold and is the second most influential factor for income.

Appendix A.1 shows that the interest rate significantly impacts farmers' ability to afford wells and influences their
income more than risk aversion and discount rate. This contrasts Figure 8, which shows that all three input factors



are equally affecting well uptake, and that risk aversion and discount rate are more important for income. This
likely stems from the sensitivity analysis parameters, where the change in interest rate is based on a factor
multiplied by the agent's initial rate, leading to minimal variation if the initial value is low. Furthermore, agents
with higher initial interest rates are already not adapting (Appendix A.1), thus are only sensitive to (one-way)
decreasing interest changes.
**4 Discussion and recommendations**
In this study, we further developed a large-scale socio-hydrological ABM to assess the adaptive responses of
different farmer agents under consecutive droughts. We show that farmers with more financial resources invest in
irrigation quickly, when a drought occurs, whereas farmers with less resources switch to less water intensive crops
to increase yields (T. Birkenholtz, 2009; T. L. Birkenholtz, 2015; Fishman et al., 2017). After the first drought, as
risk perception is still high, and income had increased, well uptake also increased among farmers with less financial
resources. In the short term, this increased the area's income and resilience, reflected in rising yields and income
over consecutive droughts. However, similar to reservoir supply-demand cycles (di Baldassarre et al., 2018), the
widespread adoption of wells led to an increase in water-intensive crops and growing of crops during the dry
season, which in turn raised water demand. During wet periods the available groundwater could support this
demand, but during dry periods the groundwater rapidly declined. Consequently, despite being less severe than
the first, the last drought resulted in many wells drying up quickly and yields declining. Furthermore, homogeneous
cultivation as a result of economic maximization made the region more sensitive to market price shocks. This was
seen from 2013 to 2015, where crop market prices of the main cultivated crops dropped, which led to a much
larger drop in farmers' average income compared to the no-adaptation scenario. Thus, although initially drought
vulnerability decreased and incomes rose, ultimately, farmer's adaptive responses under consecutive droughts
increased drought vulnerability and impact. This underscores the importance of considering consecutive events,
as focusing solely on the first event would overlook the ultimate impact. Suggested policies to address groundwater
decline and well drying while maintaining higher incomes include promoting efficient irrigation technologies
(Narayanamoorthy, 2004), implementing fixed water use ceilings (Suhag, 2016), encouraging rainwater harvesting
(Glendenning et al., 2012) or combinations of all (Wens et al., 2022).

The maladaptive path of tubewell irrigation expansion, growth of water-intensive crops, the subsequent rapid
depletion of groundwater and resulting economic decline we simulated here has been commonly observed in India
(Roy & Shah, 2002). Previous studies modelling the economics of wells show the income and groundwater
fluctuations from wells and crop changes occurring gradually (Robert et al., 2018; Sayre & Taraz, 2019). Aside
from investment costs, they show profits and groundwater levels rising and falling gradually over time, with the
simulations never experiencing shocks. However, we here observe that this is not a steady process, but rather one
characterized by periods of stabilization and rapid reduction of groundwater levels and incomes during wet and
dry periods. Additionally, under consecutive droughts, we see social- (i.e. continued loan payments, crop price
drops) and ecological shocks (i.e. lower groundwater levels, drought) coinciding (Folke et al., 2010). Therefore,
agricultural decline as described by Roy & Shah (2002) may occur more sudden and rapidly in a socio-hydrological
systems approach than what previous studies predict (Manning & Suter, 2016; Robert et al., 2018; Sayre & Taraz,
2019). Such sudden shocks are harder to adapt to, potentially leading to more severe impacts or disasters



(Rockström, 2003). Thus, for future analyses, we recommend transitioning to similar coupled agent-based
hydrological models, combined with climate data, to identify areas where drought risk is or will be high.

We also observed that adaptive patterns are spatiotemporally heterogeneous. For example, the farmers' location
determined the number of wells that could be held before depleting groundwater levels, influenced by factors like
precipitation and agent density. Water availability, resulting from precipitation and irrigation, along with market
dynamics, influenced crop choices, leading to varied cropping patterns as prices fluctuated, between wet and dry
periods, seasons, and locations upstream or downstream. Furthermore, at individual scale, we observed that
variations in farm size, access to credit, time preferences, or risk attitudes influenced farmers' adaptation decisions.
Building on our demonstration of the impact of varying hydroclimatic conditions and farmer characteristics on
adaptation behavior, and the substantial effects of this behavior on a river basin's hydrology, we again highlight
the value of large-scale coupled socio-hydrological models. These models can further enhance understanding of
both basin hydrology and farmer behavior. This is needed to design policies such that they, for example, minimize
overall impacts and specifically reduce impacts on smallholder farmers (Wens et al., 2022). By further exploiting
our methods, it is possible to attempt to identify policies that can slow the expansion of wells in areas where it is
unsustainable, while simultaneously avoiding interference in regions where growth is more sustainable, which is
recommended by Roy & Shah (2002). Furthermore, it can help in determining which adaptation alternatives and
policies can decrease drought vulnerability while simultaneously being financially attractive enough to see
adaptation beyond the village scale (Fishman et al., 2017).

In this study we were able to model emergent patterns as a result of many combined small-scale processes due to
human behavior under consecutive droughts at a river basin scale and quantitatively assess their hydrological and
agricultural impacts. However, there are several challenges related to our methods. First, coupled-ABMs require
many inputs such as calibration and validation data (McCulloch et al., 2022; Schrieks et al., 2021). Some of this
data was readily available, however, others such as spatial explicit longitudinal groundwater levels were not.
Additionally, other inputs such as drought loss thresholds are based off theory (Bubeck et al., 2012; Kahneman &
Tversky, 2013; Neto et al., 2023) and have not been determined for droughts. The precise levels of, e.g., well
uptake or income, depend on the reliability and precision of data inputs and can therefore vary (Robert et al., 2018).
Although the model is thoroughly calibrated, this paper concentrates on patterns, variations among farmers, places,
and scenario differences, rather than on absolute values. We recommend further research to develop detailed
regional data to improve the accuracy of large-scale ABMs, along with acquiring empirical data on behavioral
aspects to refine behavioral estimates. Second, crop switching steered the region to an extremely homogeneous
cultivation of certain crops that had substantially risen in price. Albeit a progression towards uniform cultivation
of crops has been observed under similar circumstances (Birkinshaw, 2022), the degree seen here is unlikely. We
incorporate economic decisions influenced by subjective risk behaviors into our analysis, as they were the central
focus of our study. However, other subjective behaviors exist, such as decisions influenced not by personal benefit
assessments, but by perceptions of others' beliefs, cultural norms, attitudes, or habits (Baddeley, 2010). Including
this type of behavior in future research may reduce homogeneity; however, no behavioral theory perfectly
encompasses all adaptive behavior (Schrieks et al., 2021). Therefore, we recommend keeping the SEUT, while
incorporating a market feedback, that lowers the profitability of commonly cultivated crops due to increased
cultivation costs and reduced market prices, calibrated with observed prices. Alternatively, we suggest adding a



calibrated unobserved cost factor for all crops (Yoon et al., 2024). Both modulate the profitability of crops and
reduce the modelled divergence from historical patterns. Furthermore, subsistence farming, which involves
cultivating crops for household consumption, could reduce homogeneity as well (Bisht et al., 2014; Hailegiorgis
et al., 2018. Subsistence farms cultivate more diverse crops and take up most of smallholder farmer's cultivated
area (Bisht et al., 2014. A proposed model implementation could mandate that all farmers dedicate one plot to
subsistence crops. This would limit the smallest farmers to their initial crop rotations, while larger farmers would
be free to cultivate commercial crops on their remaining land.
**5 Conclusions**
In this study, we assess the adaptive responses of heterogenous farmers under consecutive droughts at river basin
scale in the Bhima basin, India. To do so, we further developed a large-scale socio-hydrological agent-based model
(ABM) by implementing the Subjective Expected Utility Theory (SEUT) alongside heterogeneous farmer
characteristics and dynamic adaptation costs, risk experience and perceptions to realistically simulate many
individual's behavior. From the emergent patterns of all individual's behavior under consecutive droughts we were
able to assess river basin scale patterns and come to these three main conclusions.
First, farmer's adaptive responses under consecutive droughts ultimately led to higher drought
vulnerability and impact. Although farmer's switching of crops and uptake of wells initially reduced drought
vulnerability and increased incomes, subsequent crop switching to water-intensive crops and intensified cropping
patterns increased water demand. Furthermore, the homogeneous cultivation encouraged by economic
maximization made the region more sensitive to market price shocks. These findings highlight the importance of
looking at consecutive events, as focusing solely on adaptation during first events would overlook the ultimate
impact.
Second, the impacts of droughts on (groundwater irrigating) farmers are higher and can happen more
suddenly in a socio-hydrological system under realistic climate forcings compared to what just gradual numerical
economical models can predict. This is because groundwater depletion happens in periods of stabilization and
rapid reduction instead of gradually, and because ecological shocks (i.e. droughts) and social shocks (i.e. crop
price drops) can coincide to rapidly decrease farmer incomes.
Third, adaptive patterns, vulnerability, and impacts are spatially and temporally heterogeneous. Factors
such as market prices, received precipitation, farmers' characteristics and neighbors, and access to irrigation
influence crop choices and adaptation strategies. This variability underscores the benefits of using large-scale
ABMs to analyze specific outcomes for different groups at different times.

This research presents the first analysis of farmer's adaptive responses under consecutive droughts using a large-
scale coupled agent-based hydrological model with realistic behavior. We emphasize the added value of employing
coupled socio-hydrological models for risk analysis or policy testing. We recommend using these models to, for
example, test policies designed to minimize overall impacts or to minimize them for smallholder farmers. Further
research could also explore alternative adaptations to wells that reduce drought vulnerability and are financially
viable enough to encourage wider adoption. Lastly, we advocate for research aimed at developing detailed regional
data to improve the accuracy of large-scale ABMs, along with acquiring empirical data on behavioral aspects to
refine behavioral estimates.



**Appendix A: Additional figures**

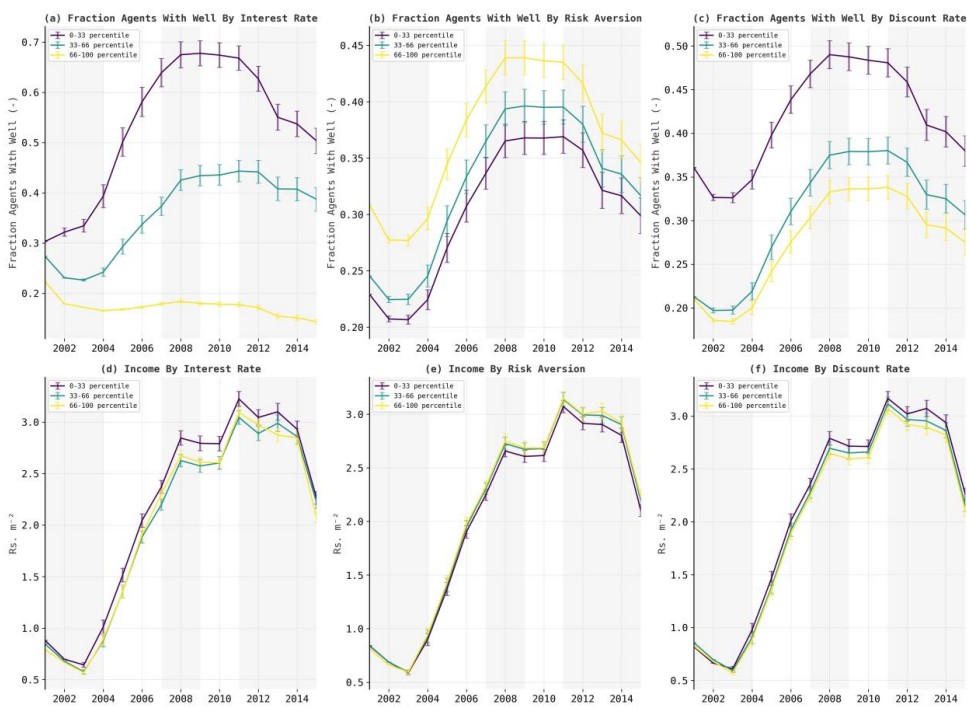


**Figure A1.** Well uptake and income grouped based on agent's interest rate, risk aversion and discount rate. The
values indicate the means of 60 runs, while the error bars indicate the standard error.



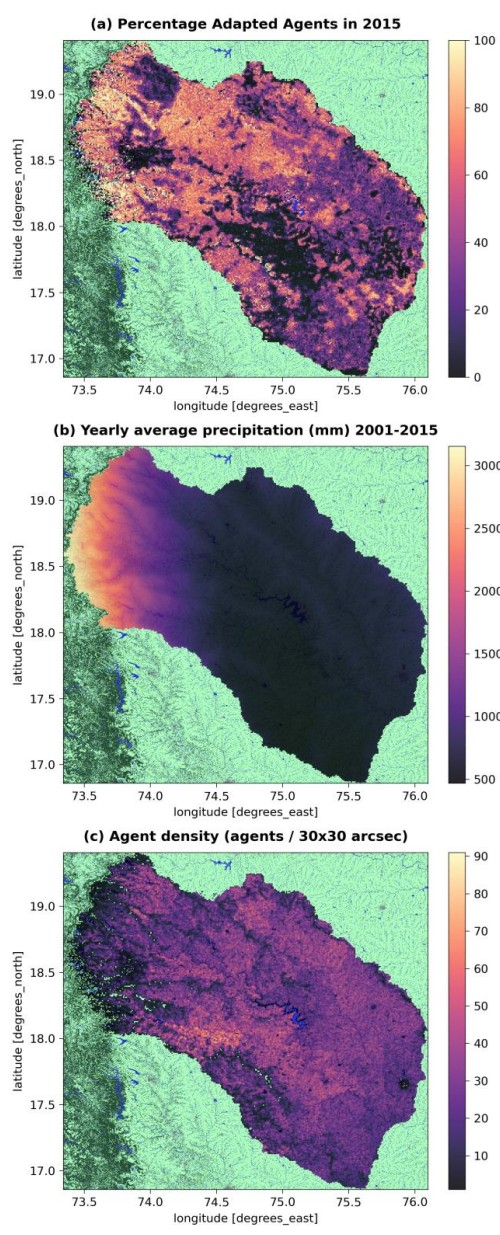


**Figure A2.** Spatial patterns of adaptation (a), precipitation (b) and agent density (c) in the Bhima basin.





**Appendix B: Model Settings & Parameters**

**Table B1.** Model settings and parametrization

| Variable / Parameter | Definition, unit | Value / range |
|---|---|---|
| **Well costs** | (Adapted from Robert et al. (2018) | |
| $C^{adapt}$ | Annual irrigation investment cost (Rs) | See B.1 |
| $D$ | Depth of Borewell (m) | Current groundwater depth + 20 m |
| $D_I$ | Initial depth of borewell of agents with well during spin-up | 42.5 m |
| $pr_D$ | Probability of well failure | 0.2 |
| *Lifespan = Loan duration (n) = Time horizon ($R_t$)* | Years | 30 |
| $C_D$ | Cost of drilling well | See B.1 |
| $C_m$ | Maintenance costs (Rs) | See B.1 |
| $W$ | Potential amount of water pumped | See B.1 |
| $FR$ | Flow rate (cubic meter per hour) | See B.1 |
| $Pr_I$ | Proportion of available water for irrigation | 1 |
| $HP$ | Pump horse power (HP) | 10 |
| $C_{HP}$ | Pump unit purchase costs (Rs) | See B.1 |
| $A_t$ | Daily power supply (hours per day) | 3.5 |
| $L$ | Total planted time (days) | Dependent on agent crop rotation, total nr of days crop is planted. |
| $C_I$ | Cost of pumping (Rs) | See B.1 |
| $E$ | Electric power used for irrigation (Rs per kilowatt hour) | See B.1 |
| $H$ | Number of hours pumping | See B.1 |
| $C_E$ | Electricity unit costs (Rs per kilowatt hour) | 0 |
| **Social parameters** | See sect. 2.3 & 2.5 | |
| $\sigma$ | Risk aversion | See sect. 2.5 Mean: 0.02; STD: 0.82. (Just & Lybbert, 2009 |
| $r$ | Discount rate | See sect. 2.5 Mean: 0.159; STD: 0.193. (Bauer et al., 2012 |



| | | |
|---|---|---|
| $r$ | Interest rate | See B.2 |
| **Risk perception** | | |
| $\beta$ | Risk perception | See sect. 2.3 for calculation |
| $c$ | Maximum overestimation of risk, calibrated | Min: 2; Max: 10; Final: **4.320833061643743** |
| $d$ | Risk reduction factor | -2.5 |
| $e$ | Minimum underestimation of risk | 0.01 |
| **Hydrological parameters (CWATM)** | (Burek et al., 2020; De Bruijn et al., 2023 | |
| SnowMeltCoef* | Snow melt coefficient. *not calibrated as no snow in study area | 0.004 |
| arnoBeta_add | | 0.14375536957497898 |
| factor_interflow | | 0.7613961217818681 |
| lakeAFactor | | 3.221318627249794 |
| lakeEvaFactor | | 2.44551165779312 |
| manningsN | | 1.3993375807912372 |
| normalStorageLimit | | 0.645563228322237 |
| preferentialFlowConstant | | 1.426435027367161 |
| recessionCoeff_factor | | 4.091720268164577 |
| soildepth_factor | | 1.7727423771361288 |
| return_fraction | | 0.44501083424619015 |
| **Calibrated parameters (ABM)** | | |
| base_management_yield_ratio | See B.3 | Min: 0.4; Max: 1; Final: **0.9942851661004738** |
| expenditure_cap | See 2.3 | Min: 0.2; Max: 0.5; Final: **0.29686828121956016** |
| drought_threshold | Drought loss threshold. See 2.3 | Min: 5; Max: 25; Final: **15.317595486070905** |
| risk_perception_max | See 2.3 | Min: 2; Max: 10; Final: **4.320833061643743** |
| **Sensitivity settings** | | |
| risk_aversion | See B.4 | Min: 0.5 Max: 0.9 |
| discount_rate | See B.4 | Min: 0.059 Max: 0.259 |
| interest_rate | See B.4 | Min: Max: |
| well_cost | See B.4 | Min norm: 0.5; Max norm: 1.5 Min: 0; Max: 1 |
| drought_threshold | See B.4 | Min: -5 |






### B.1 Well costs

*Annual investment cost*: The yearly adaptation costs are a function of the well depth, the pump's horsepower
(HP), its maintenance costs and the cost of groundwater pumping. This is adjusted for the loan duration (*n*) using
the agent's yearly interest rate (*r*).
$$C_{t,d}^{adapt} = (C_D + C_{HP}) * \frac{r*(1+r)^n}{(1+r)^n-1} + C_M + C_I$$
*Borewell construction cost:* The borewell construction cost is dependent on the probability of well failure ($pr_D$)
and the groundwater depth for the agent (D). The constants are adjusted yearly based on inflation.
$$C_D = (1 + 100 * pr_D) * (486.33 * D - 0.00824 * D^2)$$

*Initial borewell depth:* Initial borewell depth ($D_I$) of agents who had wells before the adaptation started was
based on the average groundwater depth in the Bhima basin + 20 m.
*Pump Cost:* The pump cost is dependent on the horsepower (HP) of the pump. The constant is adjusted yearly
based on inflation.
$$C_{HP} = 3570 * HP$$
*Irrigation maintenance cost:* The irrigation maintenance cost is dependent on the potential amount of water
pumped (*W*). The constant is adjusted yearly based on inflation.
$$C_M = 6598 * W^{0.16}$$
*Potential amount of water:* The potential amount of water pumped is dependent on the flow rate (FR), the total
planted time (*L*), the number of hours pumping per day ($A_t$) and the proportion of available water for pumping $pr_I$.
$$W_t = FR * L * A_t * pr_I$$
*Flow rate:* The flow rate is dependent on the groundwater table (G).
$$FR = 79.93 * G^{-0.728}$$
*Cost of groundwater pumping*: The yearly cost of groundwater irrigation ($C_I$) is dependent on the total planted
time (*L*), the number of hours pumping per day ($A_t$), the proportion of available water for pumping $pr_I$, the electric
power (E) and the electricity unit costs ($C_E$).
$$C_I = L * A_t * pr_I * E * C_E$$
*Electric power (kilowatt hour):* The electric power is dependent on the horsepower (HP) to watt conversion.
$$E = 745.7 * HP$$

### B.2 Interest rates

See section 2.5 for how interest rates were determined. The average for all farmers comes out at approximately
10.6 %, close to the observed 10.7 % of P. D. Udmale et al. (2015. Below is the table relating landholding size to
interest rate:
**Table B2.** The relation between size class and interest rate to generate interest rates for the farmer population.





| Size class (ha) | < 0.5 | 0.5-1.0 | 1.0-2.0 | 2.0-3.0 | 3.0-4.0 | 4.0-5.0 | 5.0-7.5 | 7.5-10.0 | 10.0-20.0 | > 20.0 |
|---|---|---|---|---|---|---|---|---|---|---|
| Interest rate (%) | 16 | 11.5 | 10 | 7.75 | 6.5 | 6.5 | 6.5 | 5 | 3 | 3 |


### B.3 Calibration

In addition to the parameters explained in section 2.3., there is also a base management yield ratio adjustment.
This is a parameter that shifts each agent's yield ratio with a flat rate to do a mean adjustment.

### B.4 Sensitivity

Sensitivity parameters were changed differently per parameter. The function latin.sample from SAlib (Iwanaga et
al., 2022 was used to generate 300 sets of values between the min and max. The min and max were used as inputs
to change either the absolute values of a parameter (drought loss threshold), to change the distributions of all
agent's values (risk aversion, discount rate) or change all agent's individual parameters with a fixed rate (interest
rate).
*Risk aversion:* See section 2.5 on how the initial risk aversion was determined. To change this, this distribution
was normalized and rescaled using a new standard deviation, which was a latin.sample value between the given
min and max.
*Discount rate:* Similar to risk aversion, but now instead of the standard deviation, the mean was sampled between
the min and max and used to rescale the distribution.
*Interest rate:* Each agent's individual interest rate (section 2.5, B.2) was multiplied with a sampled value between
the given min and max.
*Well cost:* The well cost factor is determined by adjusting the fixed and yearly costs by an absolute factor. This
absolute factor adjusts the price based on a normal distribution of values. The standard deviation is 0.5 (50 %
higher/lower price) and the mean is 1 (no price change). Latin.sample then samples quantile values between 0 and
1, and uses the standard deviation and mean to calculate the adjustment factor. Thus, the percentual adjustment
factor follows a normal distribution around the original price (1).
*Drought loss threshold:* An absolute value was added/subtracted from the drought loss threshold based on the
sampled values between the min and max.

### Code and data availability

The most recent version of the GEB and adapted CWatM model, as well as scripts for data
acquisition and model setup can be found on GitHub (github.com/GEB-model). The model
inputs, parametrization and code used for this manuscript are accessible through Zenodo
(Kalthof & De Bruijn, 2024). This page also includes the averages and standard deviations of
the 60 runs of the adaptation and non-adaptation scenario which are featured in all figures.



**Author contributions**

MK, JB, HDM, HK and JA did the research conceptualization; JB, HDM, HK and JA provided supervision; MK and JB MK developed the methodology and code; MK obtained and analyzed the data; MK wrote the manuscript draft; JA, JB, HDM and HK reviewed and edited the manuscript.

**Competing interests**

One of the co-authors is editor of NHESS. Furthermore, the author and several of the co-authors work at the same department of two other NHESS editors: Anne Van Loon and Philip Ward.

**Acknowledgements**

A.I. was used to assist in coding and writing.

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
