# Peer review of "Adaptive Behavior of Over a Million Individual Farmers"

_EGUsphere, 2024_

## Author Comment (AC1)

*Thank you for inviting me to comment on the manuscript: "Adaptive Behavior of Over a Million Individual Farmers Under Consecutive Droughts: A Large-Scale Agent-Based Modeling Analysis in the Bhima Basin, India". I am an expert on agent-based land use change models. This may limit my expetise on the hydrological aspects of the presented work.*

*The authors use two coupled models (one ABM and one hydrological model) GEB (please spell out what GEB stands for) to model the land use of presumably 1.4 million farmers in the face of consecutive droughts over several decades. Unfortunately, the text is very rich combing description of drought modelling, theory, ABM model, and many more things. Therefore, at least for me it is impossible to understand the model(s) in detail and to put the results into context. Therefore, I am not able to appreciate the model and its results sufficiently, although the topic is timely and I guess the approach is relevant and promising.*

*Maybe the use of a protocol such as ODD+D (Mueller et al. 2013, Describing human decisions in agent-based models - ODD+D, an extension of the ODD protocol) could help to present the model in a more digestible way. At least the authors should summarize somewhere (maybe as a table) an overview of the properties of the agents and or agent types.*

Thank you for the suggestion, this is indeed helpful as it was difficult to both cover all aspects of the model and keep a concise paper. An ODD+D has been added to the supplementary information that we hope gives better insight into the full working of the model. If you are still missing more technical insights, the paper by De Bruijn et al. (2023) discusses the technical base design of the Geographic Environmental and Behavioural (GEB) model.

*The authors present many responses from their model: "Our analysis examines how these adaptations affect profits, yields, and groundwater levels, considering, e.g., farm size, risk aversion and drought perception." Maybe it would be helpful to reduce the number of responses and/or scenarios? Especially in Figures 5 and 7 I would encourage the authors to present less panels and to focus on a more narrow narrative.*

We agree that some panels had unnecessary information and that 8 panels is too much, thus we reduced figure 5 to 3 panels and figure 7 to 6 and removed/trimmed paragraphs in the description of the results. Furthermore, we have moved the sensitivity analysis to the appendix.

However, our main finding is that the combination of hydrological and socio-economical factors steer the area and especially certain groups of farmers into a more vulnerable state. For this, combinations of the results of yield, crop choices, income, groundwater and wells are needed, as the interplay between those leads to this result. We also find

that being able to model the interplay of these factors is the major strength of this (new) type of model.

*I appreciate that the authors follow a theory to justify their decision model. However, the authors need to guide the reader more carefully, since the SEUT is not a standard theory for all Economists and certainly not for all land use change modellers. Fishburn (1981) is a review of several theories and the list of papers suggested as examples of the application of SEUT needs to be critically revised (Groeneveld (review), Haer and Wens do not mention SEUT and do not cite Fishburn). When SEUT is introduced it also should be mentioned that the authors use imitation and "bounded rationality" (line 215) as well in their decision modelling. Later on also prospect theory is considered.*

Thank you for your sharp comment. Wens (2020) compares the "economically rational" EUT with the PMT, and concludes that a more bounded rational theory covers behavior better. This was originally used as justification of using the SEUT over the EUT, but this has been lost over subsequent versions. This happened for Groeneveld as well. Haer (2020), however, does not refer to it as the SEUT, but they use the exact same theory (and calculation) of the expected utility for their boundedly rational agents in effect, where the EUT is altered by the changed perception of probabilities due to having experienced an event. The references have been changed and the justification has been expanded in the main text (line 145 to 155) and added in the ODD+D protocol (section 2.1.3).

*In general, I am missing an argument why it is useful to model > 1 million agents. Other authors decided to gain knowledge by aggregating actors to agent types in their land use models (e.g. https://landchange.imk-ifu.kit.edu/CRAFTY or the work by Millington et al. https://www.jasss.org/11/4/4.html). It would be great to see an argument developed why this computational demanding approach is seen more appropriate to answer questions of land use change. This is especially critical since the authors argue that it is not computationally feasible to compute for all agents the SEUT for all 300 options ("unique crop rotations"). Would it make sense to compute less agents and therefore consider all 300 options?*

There are several reasons why we decided to not aggregate agents (meaning for each farmer in the basin, we also simulate one agent, what we call "one-to-one"). First and foremost, we do not know what a representative agent for our area is (Page, 2012) and by pre-emptively aggregating agents, we may lose interactions that we were not aware existed in the first place (Page, 2012). This is especially true for an area so heterogeneous as the Bhima basin in India, where there are extreme differences in landholder size (Desai et al., 2008) , which factor through in other agent attributes such as which crops they initially cultivate (Department of Agriculture & Farmers Welfare India, 2001), their access to credit or their social factors (Hoda & Terway, 2015; Maertens et al., 2014; Udmale et al., 2015). For example, if we were to aggregate to one agent per grid cell, we would already lose out on the process where larger agents have more funds to invest within

similar budget constraints, and tragedies of the commons, where larger farmers extract more groundwater and adjacent smaller farmers are unable to access the deepening groundwater. Instead of one grid cell per agent, we could attempt to scale up all farmers, adaptation costs, etc. and reduce the total number of agents, but this would require many parameter scaling adjustments, and it is unknown if model processes and interactions (including those with the hydrology, for which spatial factors are generally quite important) would remain similar. We agree however that finding what results the model would produce with different levels of aggregation would be a very interesting future study, and we thus added it to recommendations (lines 557-559). However, for such research to be possible, we require the development of these large and efficient models that are able to simulate at this detail and scale in the first place.

The second reason for keeping more agents while arguing that it is inefficient to calculate 300 options for one agent, is that more agents do not scale linearly with computational times in GEB. Due to the high degree of vectorization in the model, many agents doing one operation can be simulated much more efficiently than fewer agents doing many operations (i.e., 1.5 million agents doing 1 action is substantially faster than 5000 agents doing 300 actions). Additionally, as we always need to simulate the full region's hydrology, fewer agents may not bring about the same computational advantages as with non-coupled ABMS. We mention that we do not aggregate agents (lines 111-112) and added additional clarification with regards to why we chose to not aggregate to the initialization section (lines 287 to 290) and to the ODD+D protocol (section 2.7.1).

*I have difficulties to understand the results. To my understanding imitating the strategy of more successful agents in the neighbourhood of an agent is at the heart of the presented dynamics. In the abstract this is not mentioned: "n adaptive scenarios, farmers can either do nothing, switch crops, or dig wells, based on each action's expected utility." If my reading is correct the imitation aspect should be mentioned early on and should be discussed in a diffusion of strategy/technology context. What are the updating rules – synchronous or asynchronously? How many neighbours are considered? What is the initial trait distribution of actors. Is there a spatial structure in the inital trait distribution? Is the number of farmers constant over the years? Form Figure 4 it seems that the model does not show much variation between runs. Would have been less agents sufficient? What source/help of A.I. have the authors used for what?*

Imitation in combination with calculating the utility using the SEUT is at the heart of crop switching: agents compare the expected utility of a selection of their neighbors' crop rotations and choose the rotation with the highest utility. For choosing whether to dig a well, agents do not look at their direct neighbors. However, for calculating the expected utility of doing nothing or digging a well, it is assumed agents know the "true" added value of a well. Since wells both reduce damages during drought years as well as structurally increase water availability, which affects agents differently based on, e.g., the

precipitation they receive, the crops they cultivate, etc. the "true" added value of wells is difficult to empirically predict beforehand. In other models this is often given empirically as, for example, a depth-damage curve in combination with flood maps (flooding). However, we use the agents that already had wells in the model itself to determine this added value, as we believe this gives a more accurate representation given the large differences in contexts for the agents. Therefore we don't necessarily see this as imitation of other agents, but just as a way to determine the objective added value of wells given a farmer with x crop rotation at x location. We have added more explicitly the difference between the crop switching and well adaptation to the text (161 to 171). Furthermore, the additional factors you have requested are all in the ODD+D protocol (updating rules: 1.3.1; neighbors considered: 1.3.1, 2.4.3, 2.6.3, 2.7.1; initial trait distribution: 2.1.4., added plots of distribution of initialized personal agent factors; spatial structure: 2.1.4., 2.9.1, number of farmers: line 280, 2.1.4, 3.2.1).

There is some stochasticity in the model, which was the original reason for the 60 runs (section 2.9.1 ODD+D) . It does seem this effect is rather low, and could potentially be left out for future studies. This did not have major effects on run times, as these could be run in parallel on a linux supercomputer cluster. Run times were about 20-25 hours for a parallel run and about 15-20 hours for a solo run (without spin-up).

The use of A.I. has been further clarified (lines 703-704).

*The role of the "spin-up" period (21 years) needs to be explained in more detail. The model is initialized with data from some point in time (when). Given the substantial temporal dynamics of the responses (see Figure 4) the choice of the length of the "spin up" period should have a strong effect on the results? It is written that the calibration has been done in the period after the spin up from 2001 to 2010? It is difficult for me to understand the evolution of Figure 4. The starting point at 2001 is the result of the spin up period? The period from 2001 to 2010 is calibrated and after that it is the model projection?*

We have further explained and addressed these questions about the roles and differences of the spin-up and run (line 327-341).

A full model run consists of a "spin-up" from 1980 to 2001, and a "run" from 2001 to 2015. The spin-up period serves to set-up accurate hydrological stocks in the rivers, reservoirs, groundwater etc., and to establish enough data points for the drought probability – yield relation. At the end of the spin-up, the model state is saved and used as starting point of the run. The start of the run in 2001 was chosen as both the IHDS (Desai et al., 2008) and agricultural census (Department of Agriculture & Farmers Welfare India, 2001) collected data in 2001. As the climate data was available from 1979-2016, the 12-month SPEI was available from 1980. Thus, the spin-up time between 1980 and 2001 was chosen to maximize the duration so that the drought probability-yield relation (the "objective drought risk experience") included as many drought events as possible. Adaptation only

occurs during the run. Two scenarios were run: one without adaptation, where agents maintained the same crop rotation and irrigation status as at the start of the model, and another where agents could change their crops or dig wells according to the decision rules outlined in section 2.3. Both scenarios use the same spin-up data.

The model was calibrated between 2001-2010 as we only had discharge and yield data during these years. We would have preferred to calibrate for the full climate data range (meaning until 2016) if the data were available, as the goal of this study was not prediction, but explanation about adaptation and risk under these consecutive droughts.

*How are small, medium, and large field farmers defined in terms of hectare?*

The ha cutoffs for small/medium and large farms were mentioned in the first section of results, but are now repeated in the figure descriptions for clarity.

*Thus, overall I have the impression that potentially great insights are hidden in the current text. More specific and potentially less research questions could help to narrow down the story to allow easier access to the main highlights of the study. And at least for me it would be necessary to have a clearer motivation why it is beneficial to consider so many agents at the very same time.*

**Specific comments**

*Abstract: "realistically simulate" – That is maybe personal but I would avoid phrases like "realistically simulate" since it is a model and the best one can do is to model something useful in respect to the research question.*

Agreed, it is still far off to be considered truly realistic. Removed the adverb.

*The models are written in Python?*

Yes, added it in line 104.

*Lines 73/74 What means "one-to-one scale"?*

Explained what the authors and De Bruijn et al. (2023) refer to as "one-to-one" scale: for every farmer in real life we have a representative agent.

*Lines 78: What "simple assumptions of human behaviour"?*

Expanded slightly on how behavior was represented before. Lines 77-80.

*Figure 1: it is not clear that some boxes are empty – please explain.*

This was done to signify the simplification. They are now removed. Line 102

*Line 101: reservoir operators – is this agent type considered in the presented study? "reservoir operators" are never mentioned again.*

This agent is present but not changed in this study. The ODD+D protocol discusses how the reservoir operators determine downstream and irrigation release.

*Figure 2: The land cover map presents less classes (e.g. agricultural land) than used in the results? How did the authors discriminate the different crop types?*

Section 2.5 in the main text describes how farmer agents are initialized. These are divided over the land use class "agricultural land". Each farmer controls their own hydrological response unit (HRU, added to section 1.2.4 of ODD+D) (De Bruijn et al., 2023), on which they can make their own land management decision. Thus each agent has their own plot of land where they have their own crop rotation (including the crop's specific characteristics and the implications of these for soil processes, evaporation, crop planting dates etc.).

*Lines 132ff – Can you define in term of your indicator (SPEI) what a severe, moderate and so on drought is? I guess there are thresholds? Please specify.*

Added the thresholds of the relevant categories of McKee et al. (1993) to the main text. Lines 138-140

*Line 156: Why 5 km radius – is this decision based on a sensitivity analysis?*

The 5 km was an error, it was actually 1 km, this is changed everywhere now. This was not based on any particular study, but on a general estimate on how far social networks go. A sensitivity analysis on more "meta" model settings (start date, radius, network size, etc.) would indeed be useful for future studies.

*Line 165: C_adapt is not considered in equation 4. Does it needs to read "C_input in eq. 4 on current market prices"?*

For clarity, we have changed C_adapt to C_well. The C_well signifies the costs of a well. The C_input signifies the costs of agricultural inputs (i.e. cumulative costs of seeds, fertilizers, etc.). Both signify the costs part of the equation, but indeed the line 165 was unclear, so I have added C_input there. (lines 182-184)

*Equations 1-4 please explain subscripts x, d, and m.*

Added the description in the text. Lines 182 to 187.

*Lines 197: crop costs – crop costs C_input are dependent on the type of crop? If so how?*

Yes, they are dependent on the farmers' crop type. In the preprocessing of the model, all cultivation costs are sourced from Ministry of Agriculture and Farmers Welfare in Rupees (Rs) per hectare. (https://eands.dacnet. Nic.in/Cost_of_Cultivation.htm, last access: 15 July 2022) for the full model run time. During planting, the crop that the agent is planting is taken from the crop rotation, the crop parameters are set in the model, and the cost at that specific moment in time is sourced from this premade dictionary of cultivation costs.

*Equation 9: Where is the "crop coefficient" Kc used? The duration of different harvesting stages are not crop-specific?*

All crop factors are crop specific, added clarification. The Kc is used to determine the crop-specific potential evapotranspiration from a reference evapotranspiration. Added this to the main text as well.

*"Agent initialization": Spell out IHDS, the authors should give the ranges of the agent properties. Showing boxplots or other useful representation of the distributions of attributes would be useful. I have no intuition for example how net income is initially distributed among the 1.4 million agents and how it is developing over time.*

Spelled out IHDS and added boxplots to the ODD+D in section 2.1.4.

*Line 326: (7g)?*

7g highlighted the crop market price evolution. It is now in the appendix, as it is not a model produced result, but is relevant for several figures in the results.

*Figure 4c: Is it reasonable that one crop is going from 0.05 fraction to the dominating crop?*

It is if we look at the decision rules in the model in combination with the strongly risen prices of Groundnut. However, it is not if you compare it to reality. We discuss this in the discussion, and give some explanation and recommendations to resolve it. (Lines 540 to 566). Additionally, we now added some initial argumentation in the relevant results section (lines 359-362).

*Figure 5: Very busy graph (6 panels). The authors may want to focus on some panels.*

Reduced the figure to 4 panels.

*Figure 6 and elsewhere: Specify the unit Rs*

This is the Indian currency Rupees. Added the specification.

*Figure 8: Too many panels. Legend unreadable. Please focus on the important aspects.*

Reduced the panel to 6 panels and increased the font size of legends on each figure.

*Technical corrections*

*The figure labels are too small and therefore hard to read.*

Increased the size of all legend/x/y/ labels and titles

*Figure 4c and others: colours are too similar – hard to differ crops*

The Viridis color palette was chosen to accommodate colorblind individuals. However, it is true that a continuous color palette was not the right choice for categorical data. We

now use the OKABEITO color palette (Okabe & Ito, 2002), which should still be colorblind friendly, but better suited to categorical data.

*Lines 65-66 and elsewhere: Inconsistent in-text citation of Udmale et al. 2014/2015*

Adjusted the references.

*Line 114 and elsewhere: "95 %" -> "95%"*

Replaced the instances.

*Line 263: The authors refer to figure 3 in Jun et al. 2014? Jun et al. 2014 is a comment in Nature without Figures to my understanding. Please check.*

Thank you for spotting the mistake! We intended to refer to Jun et al. 2014 to show which agricultural land data we used, and to refer to figure 2 (not 3 as we mistakenly did!) of our paper to show a visualization of the agricultural class in our study area. However, we see that this was quite unclear. We now clarified the text (Lines 318-319).

*"Sensitivity Analysis": What are 300 distinct samples. Sampling from what distribution?*

The latin.sample function from SALib uses Latin hypercube sampling. In Appendix B.4 is is further explained from what this was sampled. Lines 649-667

*Line 306: Where does stochasticity enters the model(s) and how?*

We added a description of the stochasticity in section 2.9 of the ODD+D protocol

*Citation: https://doi.org/10.5194/egusphere-2024-1588-RC1*

---

## Author Comment (AC2)

**Supplementary information 1: ODD+D protocol**

Based on the protocol by Müller et al. (2013)

**1. Overview**

**1.1 Purpose**

**1.1.1 What is the purpose of the study?**

The purpose of the study is to analyze dynamic drought risk over consecutive droughts. To do so, we use the Geographical, Environmental, and Behavioral model GEB (De Bruijn et al., 2023) . The model includes adaptive behavior of heterogeneous farmer agents that changes in response to varying hydroclimatic and socioeconomic conditions, while in turn also affecting those socio-hydrological conditions. The study is performed in the Bhima basin, India.

**1.1.2 For whom is the model designed?**

The model is designed for scientists and practitioners, particularly those interested in understanding how droughts affect adaptation of individual farmers over time and how that adaptation – in turn – affects droughts.

**1.2 Entities, state variables and scales**

**1.2.1 What kinds of entities are in the model?**

GEB includes an agent-based model (ABM) that governs the behavior of farmers and their interaction with the water cycle, as well as reservoir operators who manage water outflow from reservoirs. The ABM is coupled with a hydrological model Community Water Model (CWatM) that simulates the water cycle, availability and demand from non-agricultural sectors (e.g., domestic, energy, industry and livestock). Additionally, CWatM and the ABM are coupled to MODFLOW, which simulates the subsurface hydrology. For a full overview of CWatM and MODFLOW see (Burek et al., 2020) and (Langevin et al., 2017).

**1.2.2. By what attributes (i.e. state variables and parameters) are these entities characterized?**

**Table 1 Attributes and their values of farmer agents in GEB. "Min" and "max" refer to the minimum and maximum value used in calibration, while "Final" refers to the value resulting from the calibration process.**

| Variable / Parameter | Definition, unit | Value / range |
|---|---|---|
| Location | Where in the study area the farmers are situated (i.e., geographic coordinates of farmers) | |
| Elevation | What elevation the farm is situated at (m above sea level). | |
| Farm size | How large their farm size is. | Classes are: 'Below 0.5' acres, '0.5-1.0', '1.0-2.0', '2.0-3.0', '3.0-4.0', '4.0-5.0', '5.0-7.5', '7.5-10.0', '10.0-20.0', '20.0 & |

| | | |
|---|---|---|
| | | ABOVE'. This size is randomly generated based on the distribution of class size. After a farmer has been designated a class, an actual size is randomly generated between, e.g., 2-3 acres |
| Groundwater levels | How far below the ground the groundwater is situated. (m) below ground | Determined by MODFLOW, CWatM and groundwater extractions |
| Irrigation class | Whether farmer has used most irrigation water from groundwater, river channel or reservoirs | |
| Crop rotation | Which crops the farmer are cultivating during the Kharif, Rabi and Summer seasons | Initially determined based on the Indian Agricultural Census and Indian Human Development Survey (see de Bruijn et al., 2023) |
| Past yearly yield ratios | Array of past 20 years of average yield ratio over all seasons where farmer cultivated crops. (-) | Determined by eq. 10 sect |
| Past yearly potential and actual incomes | Array of past 20 years of potential (if no water shortage) and actual (with water shortage, determined by yield ratio) income after selling crops. (Rs) | Determined by the crop, yield ratio and market prices |
| Past yearly drought probabilities | Array of past 20 years of average Standardized Precipitation Evapotranspiration Index (SPEI) of all harvests. (-) | ~ +2 to -2 |
| Yearly costs / outstanding loan payments and durations | Yearly loan amount that farmers have to pay and how long they have to pay it for. Consists of agricultural input loans, microcredit loans and adaptation loans. | Determined by crop choice, past crop failures and well adaptation decisions. |
| **Social parameters** | See sect. 2.1.4 | |
| $\sigma$ | Risk aversion | See sect. 2.1.3 Mean: 0.02; STD: 0.82. (Just & Lybbert, 2009 |
| $r$ | Discount rate | See sect. 2.1.3 Mean: 0.159; STD: 0.193. (Bauer et al., 2012 |
| $r$ | Annual interest rate (%), coupled to land size classes. | 16, 11.5, 10, 7.75, 6.5, 6.5, 5, 3, 3 |

**Risk perception**

| | | |
|---|---|---|
| $\beta$ | Risk perception | See sect. 1.3.1 for calculation |
| $c$ | Maximum overestimation of risk, calibrated | Min: 2; Max: 10; Final: **4.32** |
| $d$ | Risk reduction factor | -2.5 |
| $e$ | Minimum underestimation of risk | 0.01 |
| $t$ | Time since last drought. (years) | 0 to the maximum runtime. |
| **Calibrated parameters** | | |
| Base yield ratio | The base yield ratio, used to adjust the mean yield ratio for calibration | Min: 0.4; Max: 1; Final: **0.99** |
| Expenditure cap | The maximum yearly costs as a fraction of income farmers can spend on loans for adaptation/inputs/etc. | Min: 0.2; Max: 0.5; Final: **0.30** |
| Drought threshold | Drought loss threshold.  See sect 2.1.3 | Min: 5; Max: 25; Final: **15.32** |
| Risk perception max $c$ | See risk perception above. | Min: 2; Max: 10; Final: **4.32** |

**Table 2 Reservoir operator agents attributes and their values in GEB**

| Variable / Parameter | Definition, unit | Value / range |
|---|---|---|
| Minimum outflow | The minimum outflow that can be set if a baseflow needs to be guaranteed (% of average discharge) | 0.0 |
| Non-damaging outflow Q | The maximum non-damaging outflow Q (% of average discharge ) | 400.0 |
| Normal Outflow Q | The normal outflow Q (% of average discharge) | 1 |
| Max reservoir release factor | Fraction of total reservoir storage to release for irrigating farmers daily (-) | Min: 0.01; Max: 0.05; Final: **0.03** |

CWatM and MODFLOW attributes. This table shows only the calibrated attributes. Full hydrological settings can be found in the CwatM.ini file on Zenodo (Kalthof & De Bruijn, 2024).

**Table 3 Calibrated CWatM attributes and their final values in GEB**

| Variable / Parameter | Definition, unit | Value / range |
|---|---|---|
| **Hydrological parameters (CWATM)** | (Burek et al., 2020; De Bruijn et al., 2023 | |
| SnowMeltCoef* | Snow melt coefficient. *not calibrated as no snow in study area | 0.004 |
| arnoBeta_add | | 0.14 |
| factor_interflow | | 0.76 |
| lakeAFactor | | 3.22 |
| lakeEvaFactor | | 2.45 |

| | |
|---|---|
| manningsN | 1.40 |
| normalStorageLimit | 0.65 |
| preferentialFlowConstant | 1.43 |
| recessionCoeff_factor | 4.09 |
| soildepth_factor | 1.77 |
| return_fraction | 0.45 |

**1.2.3. What are the exogenous factors / drivers of the model?**

The forcing data consisted of Precipitation (kg/m²/s), Surface Downwelling Longwave Radiation (W/m²), Surface Downwelling Shortwave Radiation (W/m²), Relative Humidity at Surface (%, hurs), Surface Pressure (Pa, ps), Surface Wind Speed (m/s), Near-Surface Air Temperature (K), Daily Maximum Near-Surface Air Temperature (K), Daily Minimum Near-Surface Air Temperature (K) and Wind Speed (m/s). This data was sourced from the CHELSA-W5E5 v1.0 observational climate input data at 30 arcsec horizontal and daily temporal resolution (Karger et al., 2022).

The routing was determined by identifying the outlet of the Upper Bhima basin and taking all upstream cells of it from the MERIT Hydro elevation map (Yamazaki et al., 2019), upscaled to 30″ (Eilander et al., 2021). Routing maps for river slope and width were also obtained in a similar manner (Eilander et al., 2020). Reservoir and lake footprints came from the HydroLAKES dataset (Messager et al., 2016). Where available, data on flood cushions and reservoir volumes were sourced from the Andhra Pradesh WRIMS database (https://apwrims.ap.gov.in/, last accessed on 7 September 2021). Land cover was determined from the land cover data of Jun et al. (2014).

Historical water demand is taken from CWatM and consists of domestic, industry and livestock demand following the method of Wada et al. (2011).

Crop cultivation costs are sourced from the Ministry of Agriculture and Farmers Welfare in Rupees (Rs) per hectare (https://eands.dacnet. Nic.in/Cost_of_Cultivation.htm, last access: 15 July 2022) (De Bruijn et al., 2023). Historical monthly crop market sell prices are sourced from Agmarknet (https://agmarknet.gov.in, last accessed on 27 July 2022) (De Bruijn et al., 2023) in Rupees (Rs) per kg.

**1.2.4. If applicable, how is space included in the model?**

Each field of a farmer is simulated as a single Hydrological Response Unit (HRU) (De Bruijn et al., 2023). The HRUs are dynamically sized based on the land ownership / field size of each farmer and are independently operated by each agent. This means that land management decisions such as crop rotation, planting dates and irrigation, along with soil processes like percolation, capillary rise, and evaporation, are independently simulated within a HRU for each farmer. This allows for the simulation of multiple independently operated farms within a single grid cell of CWatM (De Bruijn et al., 2023). The smallest HRU is at 30 m x 30 m, which is the resolution of the smallest cell of the land cover map.

While vertical hydrological processes like infiltration and percolation are modeled within the HRUs, river discharge and groundwater flow are handled at the grid cell level of 30″ grid size. This requires converting fluxes from HRUs to grid cells. Runoff is calculated for each HRU, aggregated based on their sizes, and then integrated into the grid cell's discharge calculations.

**1.2.5. What are the temporal and spatial resolutions and extents of the model?**

In this study, the spatial extent model is the Upper Bhima basin, but it can be configured for any region globally by selecting the appropriate outflow grid point. For the spatial resolution see section 1.2.4. The temporal extent consists of a spin-up period between 1980 and 2001, and a run period from 2001 to 2015. 2001 was chosen as there was data in 2001 for how many and which farmers had irrigation wells. As the climate data started in 1979, the 12-month SPEI was available from 1980. The spin-up time between 1980 and 2001 was chosen to maximize the duration so that the drought probability-yield relation (or the "true/objective" drought risk) included as many drought events as possible.

CWatM processes run at a daily timestep, except routing, which runs at a hourly sub-timestep. The interaction between both, such as choosing to irrigate or to harvest crops, run at a daily timestep as well. Adaptation decisions (switching crops or digging wells) are made at the end of each growing season for the next one.

**1.3 Process overview and scheduling**

**1.3.1. What entity does what, and in what order?**

*Daily timestep:* CWatM simulates all daily hydrological processes depending on, e.g., the meteorological forcing, land use types, crop potential evapotranspiration etc. Reservoir agents determine reservoir release based on the current reservoir storage and inflow plus a set minimum, normal and maximum outflow, flood cushion and maximum fill. This water is made available to be released into the river channel. The water made available for irrigating farmers is a set percentage of the total volume daily, which is first available to the most upstream agents, cascading downstream. In the same timestep, farmer agents then determine whether they will irrigate (depending on whether they have access to either reservoir water, river channel water or groundwater) and how much they will irrigate (which is determined by how much water the farmer's crop is short from fully filling field capacity so that actual evapotranspiration equals potential evapotranspiration). This water is then abstracted and added from and to the corresponding storages in CWatM, and CWatM updates all appropriate stocks, this is done asynchronously, with elevation determining the order. Each agents then checks whether it is time to plant their crops if they have no current crops planted (depending on the crop rotation of the farmer and the start of the season) and adds the input costs to their yearly costs. Additionally, they check whether it is time to harvest if they currently have a crop planted (depending on when the crop was planted and how long that specific crop grows).

[Figure]

**Figure 1 Overview of model actions, taken from** De Bruijn et al. (2023)**. The government and NGO agents do not affect the model in this paper.**

Farmers grow pearl millet, groundnut, sorghum, paddy rice, sugar cane, wheat, cotton, chickpea, maize, green gram, finger millet, sunflower and red gram. Each crop undergoes four growth stages (d1 to d4). The crop coefficient (Kc) is then calculated as follows (Fischer et al., 2021):

$$
Kc_t = \begin{cases} Kc1, & t < d_1 \\ Kc1 + (t - d1) \times \frac{Kc2 - Kc1}{d2}, & d_1 \leq t < d_2 \\ Kc2, & d_2 \leq t < d_3 \\ Kc2 + (t - (d1 + d2 + d3)) \times \frac{Kc3 - Kc2}{d4}, & \text{otherwise;} \end{cases}
$$

where $t$ represents the number of days since planting, and d1 to d4 are the durations of each growth stage. Each crop has their own set of these parameters. At the harvest stage, the actual yield (Ya) is determined based on a maximum reference yield (Yr; Siebert & Döll, 2010), the water-stress reduction factor (KyT), and the ratio of actual evapotranspiration (AET) to potential evapotranspiration (PET) throughout the growth period (Fischer et al., 2021):

$$
Y_a = Y_r \times \left( 1 - KyT \times \left( 1 - \frac{\sum_{t=0}^{t=h} AET_t}{\sum_{t=0}^{t=h} PET_t} \right) \right)
$$

After they harvest, yield is converted to income depending on the current market price of that specific crop. At the end of each season, farmers track their yield ratio of that harvest, their potential and actual profits and the 12-month SPEI of that season (from the 12-month SPEI between 1979 and 2016, calibrated from 1981-2010). They

also check whether this season's yield ratio is lower than a moving reference point plus a certain "drought threshold". The reference point is the 5-year average difference between the reference potential yield and the actual yield, and the additional drought threshold is a calibrated factor. If it is below the moving average reference point and the drought threshold (e.g., 15% below the average yield of the last 5 years), the farmer experiences a drought. In that case, their time since the last drought (table 1) resets and their risk perception rises according to

$$\beta_t = c * 1.6^{-d*t} + e$$

Where *d* is a reduction factor, *e* is a minimum underestimation of risk and c is the maximum overestimation of risk. The amount that is below the threshold is then multiplied by the yearly average income and added as a two year loan (with interest) to yearly costs as microcredit.

*Growing season / yearly timestep:* At growing season / yearly timestep the agents average those seasons' SPEI probabilities and yield ratios. The farmers are then ordered into groups of farmers that have the same crop rotation, are in the same division of the river basin (upper / middle / lower) and have wells or not. The SPEI probability and yield ratios are averaged and a relation is made between all past SPEI probabilities and yield ratios (for calculation see section 3.4), which counts as their objective risk experience (i.e. the "objective truth" of what severity/probability drought leads to what severity yield loss) (figure 2). To decide whether they will dig a well or not, they use their own objective risk experience and subjective risk perception (i.e., after a drought overestimating the probability that droughts will happen), risk aversion and discount rate to calculate the subjective expected utility (SEUT) of not adapting. As wells both increase profits during non-drought years and reduce loss during drought years, the added benefit of wells is difficult to predict. Therefore, the SEUT of wells is calculated using the objective risk experience (i.e. relation between drought probability and yield) of the same group of farmers (in the same region of the basin, with the same crop rotation) that instead do have a well, but their personal subjective risk perception, risk aversion, discount rate, interest rate for loans, and well cost (which is dependent on the local groundwater depth). For calculation see section 3.4. If the SEUT of digging a well is higher and the price of adaptation is within the farmer's budget constraints, they then adapt and the yearly loan amount (depending on well depth / cost, interest rate and loan duration, for calculation see section 3.4) is added to their yearly costs. This is done synchronously. To determine whether farmers will switch crops, all farmers calculate only their own crop rotation's SEUT and objective EUT (using neutral risk perception, aversion and discount rate). Then, agents compare their current crop rotation's SEUT with the EUT of max 5 random neighboring farmers using similar irrigation sources (within a 1 km radius, using reservoir, surface, groundwater or no irrigation). The EUT is used since using a neighbor's SEUT would mean using another agent's subjective factors. They then adopt the crop rotation of the neighbor who's EUT is highest, if this exceeds their own SEUT. This is done asynchronously, following the same order as used for irrigation.

[Figure]

**Figure 2 Specific overview of the updated behavior in this study.**

**2. Design concepts**

**2.1 Theoretical and Empirical Background**

**2.1.1 Which general concepts, theories or hypotheses are underlying the model's design at the system level or at the level(s) of the submodel(s)? What is the link to complexity and the purpose of the model?**

The modelling approach in GEB is based on a quantitative socio-hydrology framework. In this framework, we assume two-way feedback between humans and the hydrological cycle, i.e. farmers both affect and are affected by the physical (drought) environment, but also between humans and economic factors such as changing market crop prices. Furthermore, the agent-based nature of GEB acknowledges the heterogeneity of actual farmers and attempts to capture this by varying social and physical factors to produce farmer agents that are similar to the ones we see in real life.

**2.1.2 On what assumptions is/are the agents' decision model(s) based?**

Agents are boundedly rational and use the subjective expected utility (SEUT) (Savage, 1954) to choose between actions they can take. They are further influenced by the adaptive choices of their neighbors, or "imitation" (source) and by elements of prospect theory (Kahneman & Tversky, 2013; Neto et al., 2023).

**2.1.3 Why are certain decision models chosen?**

The SEUT builds on the EUT (Von Neumann & Morgenstern, 1947), by incorporating the concept of "bounded rationality", where agents remain rational utility maximizers but base their decisions on subjective estimates of drought probability. Their subjective estimates overestimate probabilities following a drought and underestimate probabilities after periods of no drought. Such boundedly rational behavior, observed in reality (Aerts et al., 2018; Kunreuther, 1996), aligns more closely with actual adaptation behavior than fully rational models (Haer et al., 2020; Wens et al., 2020). As the model's application interest is in consecutive (drought) events, this behavioral theory fit our research goals best.

However, literature indicates that human adaptive behavior is also influenced by social factors (Baddeley, 2010; Haer et al., 2016). Thus, agents also make decisions influenced by the (earlier) adaptive decisions and behavior of their neighbors. Lastly, farmers do not necessarily experience a meteorological drought as a drought, but experience drought when they experience crop loss, which is a factor of the meteorological drought, crop choice and irrigation capabilities (Van Loon et al., 2016). Furthermore, farmers also do not judge crop loss as a drought based on whether they have achieved the theoretical maximum yield if they never achieve this. Thus, we set that they only experience a drought if they have a loss against their expected gain or reference point, i.e., if the last 5 years they had on average 60% of total yield, they will experience loss if it is below this 60%. This is based on how people experience loss which is described by elements of prospect theory (Kahneman & Tversky, 2013; Neto et al., 2023). The moving reference point can change based on farmer's changed situation, e.g., if the farmer now uses irrigation and gets higher yields, if there has not been a drought for some time or if there has been a drought for a longer time (Neto et al., 2023) and yields were higher or if the farmer now has more drought resistant crops.

**2.1.4 If the model / a submodel (e.g. the decision model) is based on empirical data, where does the data come from?**

*Agent initialization:* To generate heterogeneous farmer plots and agents with characteristics statistically similar to those observed within the Bhima basin, factors from the India Human Development Survey (IHDS, Desai et al., 2008), such as agricultural net income, farm size, irrigation type or household size, were combined with Agricultural census data (Department of Agriculture & Farmers Welfare India, 2001). For this, we use the iterative proportional fitting algorithm, which reweights IHDS survey data such that it fits the distribution of crop types, farm sizes and irrigation status at sub-district level reported in the Agricultural Census (De Bruijn et al., 2023). The farmer agents and their plots were randomly distributed over their respective sub-districts on land designated as agricultural land (Jun et al., 2014) at 1.5″ resolution (50 meter at the equator). There were a total of 1432923 agents. The number of agents remained constant over the simulation period.

*Risk aversion & discount rate:* To set risk aversion and discount rate, we first normalized the distribution of agricultural net income. Then, as risk aversion and discount rate correlate with household income (Bauer et al., 2012; Just & Lybbert, 2009; Maertens et al., 2014), we rescaled the normalized income distribution with the mean and standard deviation of the (marginal) risk aversion $\sigma$ (0.02, 0.82; Just & Lybbert, 2009) and discount rate $r$ (0.159, 0.193; Bauer et al.2012) of Indian farmers. Noise was added to both to prevent that each present-biased agent is also risk taking by definition.

*Interest rates:* To account for the variation in access to credit and interest rates among farmers, we assigned each agent an interest rate based on their total landholding size, with smaller farmers receiving higher and larger farmers lower rates (Table 4, Maertens et al., 2014; P. D. Udmale et al., 2015). This is based on the interest rates observed among Indian farmers (Hoda & Terway, 2015; Udmale et al., 2015). The average for all farmers comes out at approximately 10.6%, near the observed 10.7% of (Udmale et al., 2015). Below is the table relating landholding size to interest rate:

**Table 4 Interest rates per landholding size**

| Size class (ha) | < 0.5 | 0.5- 1.0 | 1.0- 2.0 | 2.0- 3.0 | 3.0- 4.0 | 4.0- 5.0 | 5.0- 7.5 | 7.5- 10.0 | 10.0- 20.0 | > 20.0 |
|---|---|---|---|---|---|---|---|---|---|---|
| Interest rate (%) | 16 | 11.5 | 10 | 7.75 | 6.5 | 6.5 | 6.5 | 5 | 3 | 3 |

[Figure]

**Figure 3 Distributions of the farm sizes, risk aversion, discount and interest rates.**

*Calibration*: We calibrated the model from 2001 to 2010 using observed daily discharge data and yield data. The daily discharge data was obtained from 5 discharge stations at various locations in the Bhima Basin

from India-WRIS (https://indiawris.gov.in/wris/#/) . The yield data was obtained by dividing the total production by the total cropped area from (ICRISAT, 2015) to determine yield in tons per hectare. This figure was then divided by the reference maximum yield in tons per hectare to calculate the percentage of maximum yield.

*Crop market prices*: Cultivation costs which include expenses such as purchasing seeds, manure, and labor are sourced from the Ministry of Agriculture and Farmers Welfare in Rupees (Rs) per hectare (https://eands.dacnet. Nic.in/Cost_of_Cultivation.htm, last access: 15 July 2022) (De Bruijn et al., 2023). Historical monthly market prices are sourced from Agmarknet (https://agmarknet.gov.in, last accessed on 27 July 2022) (De Bruijn et al., 2023) in Rupees (Rs) per kg.

**2.1.5 At which level of aggregation were the data available?**

The IHDS is reported at household level (Desai et al., 2008), the agricultural census data available at the sub-district level (Department of Agriculture & Farmers Welfare India, 2001) and the ICRISAT meso-level database are available at the sub-district level yearly (ICRISAT, 2015). Just & Lybbert (2009) and Bauer et al. (2012) were field study experiments done at the village level in Maharastra and Karnataka, respectively. Interest rates were at the national level (Hoda & Terway, 2015).

**2.2 Individual decision making**

**2.2.1 What are the subjects and objects of decision-making? On which level of aggregation is decision-making modeled? Are multiple levels of decision making included?**

Farmers make decisions between whether to change their crop rotation or stay with the same rotation, or whether to dig a well or do nothing.

**2.2.2 What is the basic rationality behind agents' decision-making in the model? Do agents pursue an explicit objective or have other success criteria?**

Households want to maximize their subjective expected utility. This is dependent on the effectiveness of wells given their crop yield & income and drought relation, which is affected by their crop rotation and past water availability. Furthermore, this is dependent on the yearly costs of wells, which is determined by the groundwater depth at their location and their interest rate.

**2.2.3 How do agents make their decisions?**

Agents make decisions using the Subjective Expected Utility Theory, weighing the expected utility of digging a well against not digging one, and choosing the option that offers the highest utility. For crop choices, they compare the expected utility of their current crop rotation with that of their neighbors' (with similar irrigation status), selecting the crop rotation that maximizes utility.

**2.2.4 Do the agents adapt their behavior to changing endogenous and exogenous state variables? And if yes, how?**

If agents switch their crops they adapt the yield – drought probability of their neighbor whom they copied. In combination with the new sell prices of their new crops, this changes the profitability of digging wells. Reversely, if a farmer digs a well, the increased water availability now changes the profitability of their crops, leading to different behavior. If loans are taken to adapt, the yearly costs change and thus the budget constraints are tighter.

If total groundwater abstraction rises (either due to more water hungry crops or more agents with wells), groundwater levels decline and well prices rise and vice versa, leading to different behavior. Furthermore, if groundwater levels decline substantially, farmer's wells can dry, which can lead to different yields and crop decisions. If more upstream agents abstract reservoir or channel water, there is less availability for downstream farmers, leading to different profits and behavior. If crop market prices change (either input or sell costs), their profitability changes, which can lead to different crop choices.

If several years of increased yields follow, the drought reference level increases, leading to a different frequency of drought loss threshold exceedance and vice versa. Similarly, risk perception rises and falls in response to drought threshold exceedance (i.e. after a drought), leading to different behavior.

If droughts occur, incomes go down and loans go up and budget constraint change. Vice versa for wet periods. If over a longer period of time droughts become more or less frequent or severe, the yield-drought probability relation changes and behavior changes.

**2.2.5 Do social norms or cultural values play a role in the decision-making process?**

No.

**2.2.6 Do spatial aspects play a role in the decision process?**

Climatic factors depend on the location of the farmer, the number of neighboring farmers / farmer density depends on the location (a higher density downstream and a lower density upstream), both these factors and the topography and hydrology determine the groundwater depth, which is thus also dependent on the location of the farmer.

**2.2.7 Do temporal aspects play a role in the decision process?**

Farmers calculate the subjective time-discounted expected utility of each behavioral strategy by applying a decision horizon and a time discounting factor. Furthermore, the farmers have a "memory" of the past 20 years of yields and droughts, and use this to make predictions for future investments (through using the probability-yield relation in the expected utility). Lastly, just after a drought has been experienced, farmers risk perception is higher, and this goes down after time without a drought.

**2.2.8 To which extent and how is uncertainty included in the agents' decision rules?**

When farmers who do not have a well are grouped based on similarity and check their similar other farmer group that has adapted for how much yield gain per drought a well gives, a probability is given to the yield increase if the second group is much smaller than the first (e.g., the first has 100 farmers and the second only 5). The bigger the difference, the lower the probability. The probability becomes 100% if the groups are of equal size or the adapted group is bigger. If the probability estimation fails, we assume no added benefit. This is to prevent very few farmers from changing a much larger group (e.g., 3 farmers who have exceptional added benefit of wells cause 400 farmers to adapt).

When searching for neighbors with similar irrigation status (reservoir, channel or groundwater), a random selection of neighbors is taken.

**2.3 Learning**

**2.3.1 Is individual learning included in the decision process? How do individuals change their decision rules over time as consequence of their experience?**

Experience with droughts increases their risk experience and modifies their expected utility calculation and behavior. Second, through the "drought probability vs yield"-relation all farmers learn. This is both when more or less severe droughts are recorded, and the relation changes, but also when for example the farmers switches to a different crop rotation or suddenly has more water available.

**2.3.2 Is collective learning implemented in the model?**

The drought probability – yield relation is calculated of averaged values of larger groups that are similar in terms of having wells, basin location and crop rotation. This grouping is done to get more robust relations for the "objective" relation between drought probability and yield. However, one farmer may have experienced slightly different past precipitation, and will thus learn from others' drought experiences.

**2.3.3 Is collective learning implemented in the model?**

**2.4 Individual sensing**

**2.4.1 What endogenous and exogenous state variables are individuals assumed to sense and consider in their decisions? Is the sensing process erroneous?**

Farmers sense groundwater depths, well costs, irrigation class, risk perception, interest rates non-erroneous. Farmers sense future drought risk and yield, future benefits from wells and the expected utility of their own and neighbor's (with similar irrigation status) crop rotation erroneously, as these are estimations based on past and partial (risk) information and also depend on future exogeneous state variables.

**2.4.2 What state variables of which other individuals can an individual perceive? Is the sensing process erroneous?**

Farmers can sense the expected utility of other farmers with similar irrigation status's crop rotation. As this process is erroneous for the farmer itself, it is also erroneous for the farmer sensing that of the other farmer.

**2.4.3 What is the spatial scale of sensing?**

The relation between drought probability and yield is made for groups formed in the upper, middle and lower basin (regional). The expected utility of neighbors with similar irrigation status (reservoir, channel or groundwater) is within a 1 km radius / local. All other sensing is done at the farmer's own location.

**2.4.4 Are the mechanisms by which agents obtain information modeled explicitly, or are individuals simply assumed to know these variables?**

The drought probability – yield relation, well costs and expected utilities are modelled explicitly, while the other factors are just known by farmers.

**2.4.5 Are costs for cognition and costs for gathering information included in the model?**

No

**2.5 Individual prediction**

**2.5.1 Which data does the agent use to predict future conditions?**

Farmer agents use the last 20 years of drought probability and yield relation plus their risk perception, adaptation costs and crop market prices to predict future conditions.

**2.5.2 Might agents be erroneous in the prediction process, and how is it implemented?**

Yes. The drought probability-yield relation is based on the SPEI between 1979 and 2016 and events that happened to the farmer for the last 20 years and is assumed to be the fully "objective" drought risk. It thus does not know what their yield would be for an, e.g., severe drought that was not present in those last 20 years, or what a drought SPEI would be if it was not present during the 1979-2016 data series. Furthermore, when determining the added benefit of wells, it is assumed that they will remain able to extract water indefinitely, which can be changed if groundwater levels drop due to weather conditions or other agents extracting. Similarly, agents are not aware of future market price changes.

**2.6 Interaction**

**2.6.1 Are interactions among agents and entities assumed as direct or indirect?**

Direct among agents through neighbor expected utility sharing and through calculating the drought probability-yield relation. Indirect through accessing shared irrigation resources, i.e., groundwater, channel water and reservoir water.

**2.6.2 On what do the interactions depend?**

Whether farmers are similar (drought probability-yield relations) and their spatial location (upstream/downstream, many agents abstracting groundwater or relatively few, inside neighbor radius or outside).

**2.6.3 If the interactions involve communication, how are such communications represented?**

Communication for crop switching is limited to a spatial radius of 1 km of neighboring farmers with similar irrigation status (reservoir, channel or groundwater), from which a random group of 5 farmers is selected. For determining the drought probability-yield relation the communication is limited to farmers that are similar in terms of well status, basin location and crop rotation.

**2.6.4 If a coordination network exists, how does it affect the agent behaviour? Is the structure of the network imposed or emergent?**

The group of neighbors to which farmers compare the expected utility of their own crop rotation is randomized each year.

**2.7 Collectives**

**2.7.1 Do the individuals form or belong to aggregations that affect, and are affected by, the individuals? Are these aggregations imposed by the modeller or do they emerge during the simulation?**

For every farmer in the Bhima basin, we model one farmer agent (or "one-to-one" scale), thus there is no initial aggregation of agents. We do this first and foremost because we do not know what a representative agent for our area is (Page, 2012) and by pre-emptively aggregating agents, we may lose interactions that we were not aware existed in the first place (Page, 2012). This is especially true in an area as heterogeneous as the Bhima basin in India, where there are extreme differences in landholder size (Desai et al., 2008), which factor through in other agent attributes such as which crops they initially cultivate (Department of Agriculture & Farmers Welfare India, 2001), their access to credit or their social factors (Hoda & Terway, 2015; Maertens et al., 2014; Udmale et al., 2015). Aggregating while coupling to a hydrological model may also give additional issues. For example, without aggregation, if a small farmer HRU is next to a larger farmer HRU, but share the same modflow cell, they directly experience the additional groundwater decline as a result of the larger farmer extracting. If agents were aggregated and scaled, cells of groundwater would need to be crossed by the water before the decline affects each adjacent farmer. Furthermore, the idea of "representative individuals" is in itself disputed and aggregating agents, even if they are all rational utility maximizers, can lead to wrong conclusions (Axtell & Farmer, 2022; Kirman, 1992). Secondly, the vectorized design of GEB allows us to simulate more agents with only a relatively low increase in computational demand. Lastly, although it is not researched whether this has benefited the current analysis, the first step to determine the effects of aggregation is ensuring that modelling at the highest detail is possible.

During the model run, farmers are aggregated into groups that are similar in terms of well status, basin location and crop rotation. The yearly values of the drought probability and yield of those groups are averaged to determine the drought probability – yield relation. These aggregations are initially imposed by the modeler, but change throughout the simulation as agent's well status and crop rotation changes. For comparing the expected utility of farmer's current crop rotation and that of potential different farmer, neighboring farmers with similar irrigation status within a spatial radius of 1 km are searched, from which a random group of max 5 farmers is selected and the expected utilities are compared.

**2.7.2 How are collectives represented?**

As a property based on a predefined combination of traits (well status, basin location and crop rotation).

**2.8 Heterogeneity**

**2.8.1 Are the agents heterogeneous? If yes, which state variables and/or processes differ between the agents?**

Agents are heterogeneous in spatial location, which affects climatic conditions and hydrological conditions (proximity to a river or reservoir, reservoir size, upstream or downstream, groundwater depth). Agents are also heterogeneous in socio-economic conditions: they vary in accessible interest rates, discount rate and risk aversion. Farmers are heterogeneous in agricultural conditions: there are 300 different unique crop rotations, different farm sizes, differences in initial well status.

**2.8.2 Are the agents heterogeneous in their decision-making? If yes, which decision models or decision objects differ between the agents?**

Although the objective risk experience is made by aggregating the farmers into groups based on similarities, adaptation decisions are made on a personal level with all their personal characteristics (which are also dependent on earlier personal decisions, such as the yearly loan costs).

**2.9 Stochasticity**

**2.9.1 What processes (including initialization) are modeled by assuming they are random or partly random?**

When farmers who do not have a well are grouped based on similarity and check their similar other farmer group that has adapted for how much yield gain per drought a well gives, a probability is given to the yield increase if the second group is much smaller than the first (e.g., the first has 100 farmers and the second only 5). The bigger the difference, the lower the probability. . If the probability estimation fails, we assume no added benefit. This is to prevent very few farmers from changing a much larger group (e.g., 3 farmers who have exceptional added benefit of wells cause 400 farmers to adapt).

When searching for neighbors with similar irrigation status (reservoir, channel or groundwater), a random selection of neighbors is taken from the found group each time. To account for stochasticity, the model had been run 60 times and the averages of these runs have been taken.

During initialization, the farmer agents and their plots are randomly distributed over their respective sub-districts on land designated as agricultural land, which is based on the maps of Jun et al. (2014).

**2.10 Observation**

**2.10.1 What data are collected from the ABM for testing, understanding, and analyzing it, and how and when are they collected?**

GEB includes the options to report daily, monthly or yearly on all parameters. These are exported as arrays of values of the farmers. These are how much each farmer irrigates per source, their elevation, which crops they cultivate during which seasons, the groundwater depth at their location, the monthly 12-month SPEI, risk perception, yearly income, yearly yield ratio, loan payments, whether they are adapted, discharge at multiple hydrological stations, precipitation, reservoirs storages.

**2.10.2 What key results, outputs or characteristics of the model are emerging from the individuals? (Emergence)**

See the main text for the analysis.

**3.   Details**

**3.1 Implementation details**

**3.1.1 How has the model been implemented?**

Python 3 is used to implement the model, incorporating compiled Python libraries like NumPy (Harris et al., 2020) and Numba (Lam et al., 2015) for computationally intensive parts. Additionally, it features optional GPU vectorization of soil components via CuPy.

**3.1.2 Is the model accessible and if so where?**

The most recent version of the GEB and adapted CWatM model, as well as scripts for data acquisition and model setup can be found on GitHub (github.com/GEB-model). The model inputs, parametrization and code used for this manuscript are accessible through Zenodo (Kalthof & De Bruijn, 2024). This page also includes the averages and standard deviations of the 60 runs of the adaptation and non-adaptation scenario which are featured in all figures.

**3.2 Initialization**

**3.2.1 What is the initial state of the model world, i.e. at time t=0 of a simulation run?**

At the start of the spin-up in 1980 there are 1432923 farmers that have a personal interest rate, discount rate, risk aversion irrigation source, crop rotation, farm size, location and elevation. These agents remained constant over the simulated period. There are reservoirs with a certain capacity that is 90% filled and their reservoir command areas that determine which farmers have access, there are certain soil properties per land cover type and river routing information. During the spin-up there is no switching of crops or digging wells, but farmers are able to irrigate if they have access to an irrigation source.

At the end of this spin-up, all farmer parameters (table 1) are saved (including the past 20 years of drought probabilities and yields). In 2001, the "run" starts with the same values as where the spin-up ended, during which farmers can choose to dig a well or change their crop rotation. 2001 was chosen as there was data in 2001 for how many and which farmers had irrigation wells. As the climate data started in 1979, the 12-month SPEI was available from 1980. The spin-up time between 1980 and 2001 was chosen to maximize the duration so that the drought probability-yield relation (or the "true/objective" drought risk) included as many drought events as possible.

**3.2.2 Is the initialisation always the same, or is it allowed to vary among simulations?**

The initialization is always the same

**3.2.3 Are the initial values chosen arbitrarily or based on data?**

Almost all initial values are chosen based on data. See section 2.1.4. Initial groundwater levels at the spin-up were not based on data, which we would recommend for future studies.

**3.3 Input data**

**3.3.1 Does the model use input from external sources such as data files or other models to represent processes that change over time?**

Yes, see section 1.2.3.

**3.4 Submodels**

**3.4.1 What, in detail, are the submodels that represent the processes listed in 'Process overview and scheduling'?**

For a full overview of CWatM and MODFLOW see (Burek et al., 2020) and (Langevin et al., 2017).

The following submodels were not described yet in process overview and scheduling:

*Submodel expected utility calculations*:

$$SEUT_{no\_action} = \int_{p_2}^{p_1} \beta_t * p_i * U \left( \sum_{t=0}^{T} \frac{Inc_{i,x,t}}{(1+r)^t} \right) dp$$

$$SEUT_{tube\_well} = \int_{p_2}^{p_1} \beta_t * p_i * U \left( \sum_{t=0}^{T} \frac{Inc_{i,x,t}^{adapt} - C_{t,d}^{adapt}}{(1+r)^t} \right) dp$$

$$SEUT_{own\_crop\_rotation} = \int_{p_2}^{p_1} \beta_t * p_i * U \left( \sum_{t=0}^{T} \frac{Inc_{i,x,t} - C_{t,m}^{input}}{(1+r)^t} \right) dp$$

$$EUT_{own\_crop\_rotation} = \int_{p_2}^{p_1} p_i * U \left( \sum_{t=0}^{T} \frac{Inc_{i,x,t} - C_{t,m}^{input}}{(1+r)^t} \right) dp$$

Utility $U(x)$ is a function of expected income $Inc$ and potential adapted income $Inc^{adapt}$ per event $i$ and adaptation costs $C^{adapt}$. In eq. 2, $C^{adapt}$ is dependent on groundwater levels and in eq. 4 on current market prices. To calculate the utility of all decisions, we take the integral of the summed and time ($t$, years) discounted ($r$) utility under all possible events $i$ with a probability of $p_i$ and adjust $p_i$ with the subjective risk perception $\beta_t$. See table B1 for an overview of all model parameters. The utility $U(x)$ as a function of risk aversion $\sigma$ is as follows:

$$U(x) = \frac{x^{1-\sigma}}{1-\sigma}$$

*Submodel drought probability – yield calculations*:
The SPEI relation is fitted with by determining a and b in following formula, which was chosen as it they returned the highest R-squared between drought probability and yield ratio for this region (~ 0.50):

$$SPEI_{i,t} = a * log_2\left(yield_{i,t}\right) + b$$

[revised manuscript text omitted]

Eilander, D., Van Verseveld, W., Yamazaki, D., Weerts, A., Winsemius, H. C., & Ward, P. J. (2021). A hydrography upscaling method for scale-invariant parametrization of distributed hydrological models. *Hydrology and Earth System Sciences*, *25*(9), 5287–5313.

Eilander, D., Winsemius, H. C., Van Verseveld, W., Yamazaki, D., Weerts, A., & Ward, P. J. (2020). *MERIT Hydro IHU, Zenodo [data set]*.

Fischer, G., Nachtergaele, F. O., Van Velthuizen, H. T., Chiozza, F., Franceschini, G., Henry, M., Muchoney, D., & Tramberend, S. (2021). *Global agro-ecological zones v4–model documentation*. Food & Agriculture Org.

Haer, T., Botzen, W. J. W., & Aerts, J. C. J. H. (2016). The effectiveness of flood risk communication strategies and the influence of social networks-Insights from an agent-based model. *Environmental Science and Policy*, *60*, 44–52. https://doi.org/10.1016/j.envsci.2016.03.006

Haer, T., Husby, T. G., Botzen, W. J. W., & Aerts, J. C. J. H. (2020). The safe development paradox: An agent-based model for flood risk under climate change in the European Union. *Global Environmental Change*, *60*(December 2018), 102009. https://doi.org/10.1016/j.gloenvcha.2019.102009

Harris, C. R., Millman, K. J., Van Der Walt, S. J., Gommers, R., Virtanen, P., Cournapeau, D., Wieser, E., Taylor, J., Berg, S., & Smith, N. J. (2020). Array programming with NumPy. *Nature*, *585*(7825), 357–362.

Hoda, A., & Terway, P. (2015). *Credit policy for agriculture in India: An evaluation. Supporting Indian farms the smart way. Rationalising subsidies and investments for faster, inclusive and sustainable growth*. Working Paper.

ICRISAT. (2015). *Meso level data for India: 1966-2011, collected and compiled under the project on Village Dynamics in South Asia*. https://vdsa.icrisat.org/Include/document/all-apportioned-web-document.pdf

Jun, C., Ban, Y., & Li, S. (2014). Open access to Earth land-cover map. *Nature*, *514*(7523), 434.

Just, D. R., & Lybbert, T. J. (2009). Risk averters that love risk? Marginal risk aversion in comparison to a reference gamble. *American Journal of Agricultural Economics*, *91*(3), 612–626. https://doi.org/10.1111/j.1467-8276.2009.01273.x

Kahneman, D., & Tversky, A. (2013). Prospect theory: An analysis of decision under risk. In *Handbook of the fundamentals of financial decision making: Part I* (pp. 99–127). World Scientific.

Kalthof, M. W. M. L., & De Bruijn, J. (2024). *Adaptive Behavior of Over a Million Individual Farmers Under Consecutive Droughts: A Large-Scale Agent-Based Modeling Analysis in the Bhima Basin, India [Data set and Code]*. Zenodo. https://doi.org/10.5281/zenodo.11071746

Kunreuther, H. (1996). Mitigating disaster losses through insurance. *Journal of Risk and Uncertainty*, *12*, 171–187.

Lam, S. K., Pitrou, A., & Seibert, S. (2015). Numba: A llvm-based python jit compiler. *Proceedings of the Second Workshop on the LLVM Compiler Infrastructure in HPC*, 1–6.

Langevin, C. D., Hughes, J. D., Banta, E. R., Niswonger, R. G., Panday, S., & Provost, A. M. (2017). Documentation for the MODFLOW 6 Groundwater Flow Model. In *Techniques and Methods*. https://doi.org/10.3133/tm6A55

Maertens, A., Chari, A. V., & Just, D. R. (2014). Why farmers sometimes love risks: Evidence from India. *Economic Development and Cultural Change*, *62*(2), 239–274. https://doi.org/10.1086/674028

Messager, M. L., Lehner, B., Grill, G., Nedeva, I., & Schmitt, O. (2016). Estimating the volume and age of water stored in global lakes using a geo-statistical approach. *Nature Communications*, *7*(1), 13603.

Müller, B., Bohn, F., Dreßler, G., Groeneveld, J., Klassert, C., Martin, R., Schlüter, M., Schulze, J., Weise, H., & Schwarz, N. (2013). Describing human decisions in agent-based models–ODD+ D, an extension of the ODD protocol. *Environmental Modelling & Software*, *48*, 37–48.

Neto, G. G. R., Kchouk, S., Melsen, L. A., Cavalcante, L., Walker, D. W., Dewulf, A., Costa, A. C., Martins, E. S. P. R., & Oel, P. R. Van. (2023). *HESS Opinions : Drought impacts as failed prospects*. 4217–4225.

Page, S. E. (2012). Aggregation in agent-based models of economies. In *Knowledge Engineering Review* (Vol. 27, Issue 2, pp. 151–162). https://doi.org/10.1017/S0269888912000112

Savage, L. J. (1954). The foundations of statistics; jon wiley and sons. *Inc.: New York, NY, USA*.

Siebert, S., & Döll, P. (2010). Quantifying blue and green virtual water contents in global crop production as well as potential production losses without irrigation. *Journal of Hydrology*, *384*(3–4), 198–217.

Udmale, P., Ichikawa, Y., Manandhar, S., Ishidaira, H., Kiem, A. S., Shaowei, N., & Panda, S. N. (2015). How did the 2012 drought affect rural livelihoods in vulnerable areas? Empirical evidence from India. *International Journal of Disaster Risk Reduction*, *13*, 454–469. https://doi.org/10.1016/j.ijdrr.2015.08.002

Van Loon, A. F., Gleeson, T., Clark, J., Van Dijk, A. I. J. M., Stahl, K., Hannaford, J., Di Baldassarre, G., Teuling, A. J., Tallaksen, L. M., Uijlenhoet, R., Hannah, D. M., Sheffield, J., Svoboda, M., Verbeiren, B., Wagener, T., Rangecroft, S., Wanders, N., & Van Lanen, H. A. J. (2016). Drought in the Anthropocene. *Nature Geoscience*, *9*(2), 89–91. https://doi.org/10.1038/ngeo2646

Von Neumann, J., & Morgenstern, O. (1947). *Theory of games and economic behavior, 2nd rev*.

Wada, Y., Van Beek, L. P. H., & Bierkens, M. F. P. (2011). Modelling global water stress of the recent past: On the relative importance of trends in water demand and climate variability. *Hydrology and Earth System Sciences*, *15*(12), 3785–3808. https://doi.org/10.5194/hess-15-3785-2011

Wens, M., Veldkamp, T. I. E., Mwangi, M., Johnson, J. M., Lasage, R., Haer, T., & Aerts, J. C. J. H. (2020). Simulating Small-Scale Agricultural Adaptation Decisions in Response to Drought Risk: An Empirical Agent-Based Model for Semi-Arid Kenya. *Frontiers in Water*, *2*(July), 1–21. https://doi.org/10.3389/frwa.2020.00015

Yamazaki, D., Ikeshima, D., Sosa, J., Bates, P. D., Allen, G. H., & Pavelsky, T. M. (2019). MERIT Hydro: A High-Resolution Global Hydrography Map Based on Latest Topography Dataset. *Water Resources Research*, *55*(6), 5053–5073. https://doi.org/10.1029/2019WR024873

---

## Author Comment (AC3)

*Thank you for the invitation to review this manuscript. In this work, the authors extend the GEB, a coupled agent-based hydrological model, with the Subjective Expected Utility Theory and apply the model for analysis of the Bhima River basin in India under consecutive droughts. The manuscript is impressive for the complexity of model integration and the breadth of analysis conducted. I especially commend the authors for the extensive sensitivity analysis that is conducted using the model, which is often a critical gap of coupled human-water systems analyses. However, the extensiveness of the manuscript is a double-edged sword, with the manuscript very challenging to wade through given the sheer amount of material (as reviewer #1 also noted). In this sense, I reiterate reviewer #1's comments in regards focusing the analysis. I have additional comments in regards to the manuscript:*

Thank you for your positive words and overall constructive feedback! The extensiveness has indeed also been referred to by reviewer #1. Therefore, we have reduced the number of panels in figures 5 and 7, removed several paragraphs that did not contribute as much to the discussion points, and trimmed the remaining paragraphs to have a more focused narrative.

However, reiterating the response to reviewer #1, our main finding is that the combination of hydrological and socio-economic factors steer the area and especially certain groups of farmers into a more vulnerable state. For this, combinations of the results of yield, crop choices, income, groundwater and wells are needed, as the interplay between those leads to this result. We also find that being able to model the interplay of these factors is a major strength of this type of model.

*1.   My first and foremost comment is that the authors should demonstrate the validity and reasonability of the model in relation to real-world observation / understanding. While I understand that a full-scale, spatiotemporal validation of the model isn't likely possible given the sparsity of real-world observations and the complexity of the model, one can still ask the question: does the model better capture real-world patterns of the complex system in comparison to alternative approaches (e.g., the no adaptation alternative). For example, model results indicate that there is a very significant uptake in groundwater wells for large farms (growing from 30 percent of farms to 65 percent of farms) over the course of the model run. Is there any real-world quantitative or qualitative data that supports these model results? The onus in this case would be demonstrating that the adaptive model outperforms the non-adaptive model in replicating these large-scale patterns observed in reality. Similarly, do we in reality see the significant increases in groundwater depletion associated with the adaptive behavior (~10 meters in relation to the non-adaptive version); I would imagine that even apart from point groundwater level measurements, such a stark difference in depletion could be corroborated by GRACE, or even other qualitative sources. Cropping patterns are another example, the adaptive model shows large-scale crop switching that could likely be corroborated, in a broad*

*scale sense, via agricultural census information or remote sensing data. While the modeling integration and advances are impressive, there are so many choices that are made in regards to theory and implementation (as is the case with nearly all coupled human- natural models), that it becomes nearly impossible to assess the value of these model improvements in the absence of such evaluation.*

This is indeed a valuable suggestion and in the revised version of our paper, we have added a new paragraph to the discussion section where we explicitly verify the modeled trends (e.g., uptake of wells) with literature (lines 564-584). This is mainly centered around the findings of Roy & Shah (2002), which describe multiple stages in a process of well expansion and decline in many locations in India (Figure 1). Additionally, we refer to observed well uptake percentages and observed groundwater decline rates. Regarding crop choices, we discuss how our choice of behavioral theory without sufficient negative feedback effects led to too homogeneous cultivation and propose methods to simulate this more realistically.

[Figure]

Figure 20. Rise and fall of groundwater socio-ecology in India.

*Figure 1: Stages of groundwater irrigation (Roy and Shah, 2002))*

*2. As I understand, the region is also heavily managed in regards to the surface water supply system (reservoirs, diversions, manmade canals, etc.), which influences water availability for irrigation and associated demand for groundwater and farm decisions to install a groundwater well. Can the authors speak to the capabilities or limitations of CWatM in effectively representing surface water deliveries for irrigation in this region and how this may be influencing results?*

It is indeed true that the area is heavily managed, and we have therefore included several features in the model specifically to address this supply system. Yet, there are limitations and uncertainties.

First of all, reservoir command areas are included in the model. The delineation of the command areas was obtained from the India Water Resources Information System, and manually linked to reservoirs (De Bruijn et al., 2023). In principle, agents can abstract water from these reservoirs if they are in the reservoir command area and have access to the reservoir based on census data.

However, the current reservoir management module follows relative simple decision rules simulating two types of release: (a) the first is release into the river channel, which is based on protocols for reservoirs that are designed for power generation. (b) The second is a daily fixed proportion of total reservoir storage that gets released to farmer agents to abstract from. In the model, there are no physical canals delivering water to agents; instead, agents directly extract water from the reservoir as long as it remains within the daily allocated budget. Upstream agents have priority in water extraction, simulating the way canal water delivery functions in this region (see section 1.3.1. in ODD+D protocol). The volume of these releases are too low, and we, therefore, see relatively little effects of reservoirs in our results. For future research, we want to improve this module and better represent the different types of reservoirs and their effects on farmer adaptation. However, as you and reviewer #1 have both remarked, there are already many elements in the manuscript, thus we have decided to leave it out of the main text, and left the reservoir agents descriptions in the ODD+D protocol.

*3. In this discussion, the authors note that groundwater well drilling is potentially maladaptive, as farmers then rely on wells that can go dry during subsequent droughts. These are important findings that seem to be largely glossed over in the results section. For example, there isn't a figure reporting on the drying of these wells during subsequent droughts.*

Thank you for noting this statement was not properly linked to the presented results. The drying of wells is represented by the trend of the well percentage (Figure 7; particularly the 2011-2015). These figures refer to wells *with* groundwater access, or "wet" wells. However, to improve the communication of our findings we renamed the figures and changed the descriptions to make this more clear. This downward trend can be due to the

effect of wells not being replaced after their maximum lifespan was exceeded, and the drying of wells. However, the drop is too large to be fully explained by the non-replacement and it coincides with the groundwater decline, thus we can attribute the drop in well uptake fraction mainly to drying wells. While we feel adding an additional figure would increase the amount of material again – which both reviewers noted that we should avoid – we have included these notes much more explicitly in the descriptions of the results (section 3.1).

4. *It would seem to me that the imitation technique (described in lines 155-156) would very quickly lead to homogenization of crops across farmers using the same irrigation technology. Is this not the case? Could the authors further comment?*

This is indeed a very relevant remark, We include imitation together with the SEUT as this is how adaptation has been observed to spread in real life (Baddeley, 2010). Thus imitation in itself would not directly lead to homogenization. However, we agree that in our case, there is indeed too much homogenization. In the revised version of the paper, we discuss several reasons for this feature in the discussion section. First, there is an absence of economic feedbacks. This is especially important since the economic behavior theory we have implemented is mainly based on utility maximization, thus it would require feedbacks in the same domain. Second, there is no accounting of other factors influencing crop choice, such as cultural factors, intention to behavior gaps, unobserved cost factors (similar to Yoon et al. (2024)), and e.g. the prevalence of subsistence farming in the area. While it's true that once a crop rotation option is eliminated it can no longer be chosen, leading to homogenization, we believe that this elimination itself isn't inherently negative. However, the mechanisms driving it should be modeled more realistically.

We have included recommendations to improve methods and future studies can incorporate either additional economic feedbacks, such as a crop market, ensuring that farmer profits go down as more farmers grow a particular crop. However, due to the already complex methodology, we have reserved this for future work. These options are discussed in lines 584 to 599.

Additionally, reducing the number of crops and crop rotations would allow us to let agents compare the different options, without having to rely solely on imitation for computational reasons. Influence of neighbors could then be translated in an adjustment of the intention factor for example. Additionally, instead of letting agents choose between all possible crops, we may explore the decision between crop variety options, allowing agents to select varieties of their main crop that are more resistant to drought/water-efficient/etc, which also fits with literature (Drugova et al., 2021).

5. *I understand that the political economy of sugarcane is particularly influential on water security outcomes in the region (e.g.,*

*https://iopscience.iop.org/article/10.1088/1748-9326/ab9925/meta). Could the authors speak at all to how such considerations factor into the analysis? More broadly, crop prices are a significant driving factor of farm behavior, but the subjective expected utilities are only formulated in relation to subjective drought perception. Can the authors comment on whether/how farmer perceptions of economic conditions might influence results (even if outside the scope of this analysis)?*

Indeed, we refer to economic shocks coinciding with meteorological shocks, but only simulate behavior change in response to the latter. In the model, following the SEUT theory, the behavior of farmers is strongly dependent on crop prices, meaning that if prices drop, agents will start cultivating different crops that are now more profitable, and vice versa. We observe this effect for droughts in the model results, but similar behavior would be exhibited in the case of price drops or increases due to other external factors.

But on a perceptional or behavioral level you would expect that farmers fall back on crops which give security (especially during uncertain times). These could be subsistence crops for smaller farmers (mentioned in the discussion), or indeed crops such as sugarcane which have a guaranteed sale and price set by the government. This may require a second behavioral factor, which instead of reweighting the probability of future events, would reweight future crop prices based on their probabilities (e.g. 50% chance at higher prices, or 100% price of a slightly lower price) and would be similarly reweighted by risk perception and risk aversion. Perhaps this could be implemented alongside forecasts, where crops are weighted based on how well it would do in the current forecast along with the forecasts' probability weights? We included these recommendations in the revised version of our paper (lines 600 to 603)

*6.    The above article is conducted as part of the Stanford FUSE project, which was an outgrowth of the Stanford Jordan Water Project (JWP) which also introduced a coupled agent-hydrologic model for similar types of analysis (e.g., https://www.nature.com/articles/s41893-023-01177-7; https://www.pnas.org/doi/abs/10.1073/pnas.2020431118). While much of this work was focused in Jordan rather than India, these are important studies to note as part of the literature context. Can the authors speak more to how the current effort relates to and is distinguished from this line of coupled agent-hydrological model?*

Thank you for bringing the attention to these papers. There are indeed **many** similarities, and unfortunately we have only seen this research as of now. After carefully examining this literature, we think the differences lie in four areas.

- First, our research is focused on drought *events* specifically: What happens during drought events, what happens over consecutive events, how does the crop yield change (which required, for example, a more extensive crop module), how does this affect profits, etc. This event focus is also a step towards future studies where

agents have to react to alternating droughts AND floods, which is a different path compared to these studies.

- Second, is that the focus in our paper is more on the differences in situations and behavioral aspects of farmers: how do they make investment decisions using past experiences of droughts, how is this affected by a risk perception, risk aversion, time preferences, farm size, difference in climate between upstream or downstream and how do all these factors affect their choices? Some of these factors are present in the linked studies (and other spatial factors, like transportation costs are equally important but not of relevance for our study), but they are implemented slightly more rudimentary and are less the focus of the research.

- Third, we simulate all agents and localized abstractions instead of using representative agents.

- Lastly, of course we have very different local conditions which require a different model set up. For example, in our study area electricity is subsidized to cost nothing or almost nothing, which means that the costs are in the loan for the initial investment, and not in the structural price of water. This makes the differences between access in groups much more strict and leads to other dynamics which are characteristic of the area (like many agents at once losing access during a drought) and is a clear difference between these models. For future studies in the global north we do intend to make it more similar to these papers (and to Yoon et al. (2024)), where it is assumed that if agents could have gotten access to groundwater, they would already have, and now pay a price per volume of water (dependent on the groundwater levels, pumping costs, etc.) they use instead of for getting access to the water. The investment decisions are then focused on different ways of decreasing water use, like switching crops or improving irrigation.

We agree that these are good examples of similar socio-hydrological models and have added references to the socio-hydrological nature of these papers in the introduction.

*7. Figure quality throughout could be improved. Resolution is often poor with text difficult to make out and colors often hard to distinguish (e.g., couldn't distinguish crops in the cropping figs). Fig 1 is also difficult to interpret and missing text in boxes.*

This was addressed in my comments to the previous reviewer; we now use the OKABEITO color palette (Okabe & Ito, 2002), which should still be colorblind friendly, but better suited to categorical data. All labels have been made larger and figure 1 has been updated. Figure quality decreased when we exported the data to PDFs, but production quality figures (300 dpi) will be made available.

*8. Lastly, I agree with reviewer #1's comment regarding the >1 million agents. Even if such # of agents is warranted, headlining the # so prominently throughout the paper (in title, abstract, etc) in my opinion misplaces focus and potentially signals the wrong*

*message (e.g., model complexity for the sake of model complexity). This ability to model of large # of agents was already heavily featured/highlighted in the original GEB paper, so in this case I'd rather see the spotlight placed on the insights drawn from the modeling improvements and analysis, rather than the # of agents that can be modeled.*

We agree that this was the main focus of the original GEB paper, while this manuscript focusses more on the analysis and results that can be performed with such an approach. We have altered the title to reflect this, and now reads: "Adaptive Behavior of Farmers Under Consecutive Droughts Results In More Vulnerable Farmers: : A Large-Scale Agent-Based Modeling Analysis in the Bhima basin, India". Thank you for the suggestions, we do believe this is a much more fitting title.

Baddeley, M. (2010). Herding, social influence and economic decision-making: Socio-psychological and neuroscientific analyses. *Philosophical Transactions of the Royal Society B: Biological Sciences*, *365*(1538), 281–290. https://doi.org/10.1098/rstb.2009.0169

De Bruijn, J. A., Smilovic, M., Burek, P., Guillaumot, L., Wada, Y., & Aerts, J. C. J. H. (2023). GEB v0. 1: a large-scale agent-based socio-hydrological model–simulating 10 million individual farming households in a fully distributed hydrological model. *Geoscientific Model Development*, *16*(9), 2437–2454.

Drugova, T., Curtis, K. R., & Ward, R. A. (2021). Producer preferences for drought management strategies in the arid west. *Renewable Agriculture and Food Systems*. https://doi.org/10.1017/S1742170521000259

Roy, A. D., & Shah, T. (2002). Socio-ecology of groundwater irrigation in India. *Intensive Use of Groundwater Challenges and Opportunities*, 307–335.

Yoon, J., Voisin, N., Klassert, C., Thurber, T., & Xu, W. (2024). Representing farmer irrigated crop area adaptation in a large-scale hydrological model. *Hydrology and Earth System Sciences*, *28*(4), 899–916. https://doi.org/10.5194/hess-28-899-2024

---

## Author Response (AR2)

**RC1**

*Dear authors, I very much appreciate the improvements of the manuscript and think it is ready to be published. I appreciate the provision of an ODD+D. I have only checked the first equation in the ODD+D in section 3.4.1: SEUT_no_action and realized that in the ms the authors have the coefficient beta and p depending on the agent x and in the ODD+D description this has not been done. Thus, I would encourage the authors to carefully double check that everything is consistent. Apart from that I guess this article is ready to be published and hopefully will be discussed in the scientific literature.*

Thank you for your extensive first comments and for providing your second comment. We changed the equations in 3.4.1 in the ODD+D to match to those in section 2.3 of the manuscript. Furthermore, we went through both the ODD+D and the manuscript for last checks and ensured that all are consistent.

**RC2**

*Thank you to the manuscript authors for their thorough and detailed response to my initial review comments. I find the manuscript to be improved with their revisions. I also acknowledge the complexity of the modeling effort at hand, and appreciate that the authors have added material to the text acknowledging this complexity, and making clear where the analysis is necessarily limited in scope.*

*My primary remaining comment is that I found the authors response regarding model assessment/validation to be rather thin. To assess the performance of the model, the authors primarily compare model results to large-scale patterns of groundwater development/exploitation/depletion that is described in Roy and Shah (2002), and indicate that model results replicate this general pattern. While I am not intimately familiar with Roy and Shah (2002), upon my brief examination it seems to argue that groundwater use is largely driven by demand-side factors (e.g., demands for crops) and does not present a strong argument that increased groundwater exploitation is due to drought our lack of surface water availability (it seems to in fact argue against this in many instances). While the patterns of groundwater development/exploitation/depletion may nonetheless be similar (I'd imagine this could be replicated even without any drought response mechanism in place in the model), this inconsistency is rather conspicuous and weakens the assessment/validation of model performance using Roy and Shah.*

*As the model design is premised on groundwater adoption/use as a surface water drought-response mechanism, I think that identifying evidence that specifically bolsters the specific connection between groundwater development and drought response would be much more relevant to assess whether model dynamics are indeed representative of*

*reality. At the very least, I think the use of Roy and Shah to corroborate model results and dynamics should be clarified.*

Thank you for the comment. Indeed, the patterns described by Roy and Shah could likely be replicated without incorporating the drought response mechanism. Studies focused solely on well deepening, such as those by Sayre & Taraz (2019) and Robert et al. (2018), have successfully replicated this specific mechanism. However, these studies—aside from being limited to a single well in a one-dimensional scenario—are less accurate because they omit critical drought mechanisms that our study incorporates. Below we identify the four key ways in which droughts influence farmers' adaptation dynamics in our model, and show that these processes match with literature.

*Our modelling results show that*:

**1.** Drought response boost the uptake of wells through increased risk perception. This initial well uptake then boosts short-term (water/drought) resilience. However, our results and sensitivity analysis also show that economic factors (i.e. interest rates and well cost) are most important for well uptake, as risk perception is higher for subsequent droughts, but is unable to boost well uptake due to higher well costs and indebtedness;

**2.** Droughts lead to overextraction from boreholes to compensate for the lack of rainfall, which results in the accelerated decline of groundwater depth and wet wells;

**3.** Droughts result in the switching to more water-resistant crops for farmers without irrigation access;

**4.** Droughts result in failed harvest and indebtedness through having to compensate crop losses with micro-credit / loans and needing to pay outstanding loans for dry wells.

*This is consistent with the following literature results*.

**1.** Shah (2009) reports that "*groundwater wells have been the principal weapon Indian farmers have used to cope with droughts*" and that "*this is evident in the fact that well digging has tended to peak during years of droughts*". It can also be seen in figure 4.1 of Pahuja et al. (2010), where we see increases the usage of groundwater sources during/after dry periods. From studies in other regions we know that varying, e.g., risk perception is a way to capture such behavior (Aerts et al., 2018; Kunreuther et al., 1985; Schrieks et al., 2021; Tierolf et al., 2023). Furthermore, Solomon & Rao (2018) report, for example, that "*groundwater usage in semi-arid regions has increased the short-term resilience of communities in the region .. however, the exploitation of the resource for irrigation has resulted in critical groundwater levels*" and that "*... monsoonal irregularity along with increasing instances of drought has prompted farmers to adapt by shifting from supplemental to complete groundwater irrigation*", and Udmale et al. (2015) report that "*The extent of irrigation played a key role in mitigating drought damage to crops ... shows the importance of bringing more crop areas under irrigation to increase farmers'*

*adaptive capacity to drought.".* However, we acknowledge that drought (risk perception) is not always the primary factor for farmers driving well adoption. Instead, demand-side considerations or cultural influences (Solomon & Rao, 2018) can play a more significant role.

**2.** "*groundwater-irrigated area in Jaipur actually declined by over 10 percent between 2001 and 2006 due to groundwater overdraft and drought in the 2005–2006 cropping season*" (Birkenholtz, 2014), "*Groundwater extraction is increasing every year, except for a partial (but temporary) recovery following years of exceptionally heavy monsoon rainfall. Excessive pumping of groundwater to cope with drought impacts has led to groundwater depletion, which is an important concern of Maharashtra State.*" (Udmale et al., 2014), "*In water-scarce years, farmers and utilities resort to groundwater to compensate for inadequate rainfall and surface water supplies.*" (Pahuja et al., 2010).

**3.** Many studies report that farmers change to low water consuming crops as drought adaptation (Fishman et al., 2017; Udmale et al., 2014) and, e.g., switch back to traditional drought-tolerant crop varieties after wells have gone dry (Birkenholtz, 2009).

**4.** Similarly, taking out loans after drought loss and difficulties repaying loans after droughts is often observed (Solomon & Rao, 2018; Udmale et al., 2014, 2015).

Furthermore, the overall sequence around wells, crop choices, debts and droughts that we observed, namely: groundwater well irrigation expansion, initial higher resilience, a shift to high-value water-intensive crops, rapid groundwater depletion due to overdraft and drought, farmer failures, rising indebtedness, agricultural decline and a subsequent return to non-commercial crops has been documented (that explicitly mention the role of droughts) by studies beyond those of Roy and Shah (Birkenholtz, 2014; Pahuja et al., 2010; Solomon & Rao, 2018).

We have changed the manuscript in the following ways:

**1.** In the introduction where we explain our choices to implement the SEUT we have added references to Shah (2009), Pahuja et al. (2010), (Solomon & Rao, 2018) and (Udmale et al., 2014, 2015) (lines 76-83) and in the discussion where we compare the pattern of well uptake and the sensitivity analysis results to literature: "*However, although we anticipated that changes in risk perception would have a stronger impact on well uptake, our results show that economic considerations were predominantly the driving factor. This aligns with other studies which mention drought response as a major driver of well uptake (Pahuja et al., 2010; Shah, 2009), but call social and economic aspirations as the main driver (Solomon & Rao, 2018).*" (lines 543-546).

**2.** In the discussion where we emphasize the effect more (lines 506-507). Furthermore, we compare this effect to observations and to previous modelling studies which only focused on demand-side factors and failed to capture this effect: "*Furthermore, it*

*provides a much better representation of the accelerated groundwater decline during droughts observed in the field (Birkenholtz, 2014; Pahuja et al., 2010; Udmale et al., 2014), which was not captured in previous well modeling studies (Robert et al., 2018; Sayre & Taraz, 2019)."* (lines 539-542).

**3.** In lines 481-482 we mention that farmers with less resources or no wells switch to more drought resistant crops; In lines 542-543 we mention that: *"our results reflect a similar pattern of crop choice observed in the field, where farmers facing water scarcity during and after droughts switch to drought-tolerant crops (T. Birkenholtz, 2009; Udmale et al., 2014)."*

4. We added references to line 508 and 538 about continued loan payments / indebtedness.

Lastly, we added the well sequence as described above with references in lines 536-539.

We agree that only using Roy & Shah (2002) was inadequate and looked critically at our own results again. We hope that the changes are sufficient. Thank you for once again having a critical yet constructive view on this research.

*As I mentioned above, I realize that the development, description, and application of a model of such complexity is necessarily limited, especially when confined to a single manuscript, and I appreciate the lengths to which authors have already gone in this work. I believe that the work is deserving of publication, but the assessment/validation attempts should be strengthened or at the very least clarified/qualified such as to improve the description and impact of the work.*

Thank you for your understanding of the work involved in such a complex model. We are thankful for your feedback and hope you deem the manuscript sufficiently improved for publication.

Sources:

Aerts, J. C. J. H., Botzen, W. J., Clarke, K. C., Cutter, S. L., Hall, J. W., Merz, B., Michel-Kerjan, E., Mysiak, J., Surminski, S., & Kunreuther, H. (2018). Integrating human behaviour dynamics into flood disaster risk assessment. *Nature Climate Change*, *8*(3), 193–199. https://doi.org/10.1038/s41558-018-0085-1

Birkenholtz, T. (2009). Irrigated landscapes, produced scarcity, and adaptive social institutions in Rajasthan, India. *Annals of the Association of American Geographers*, *99*(1), 118–137. https://doi.org/10.1080/00045600802459093

Birkenholtz, T. (2014). Knowing Climate Change: Local Social Institutions and Adaptation in Indian Groundwater Irrigation. *Professional Geographer*, *66*(3), 354–362. https://doi.org/10.1080/00330124.2013.821721

Fishman, R., Jain, M., & Kishore, A. (2017). When water runs out: Adaptation to gradual environmental change in Indian agriculture. *Available Here*.

Kunreuther, H., Sanderson, W., & Vetschera, R. (1985). A behavioral model of the adoption of protective activities. *Journal of Economic Behavior & Organization*, *6*(1), 1–15.

Pahuja, S., Tovey, C., Foster, S., & Garduno, H. (2010). *Deep Wells and Prudence: Towards Pragmatic Action for Addressing Groundwater Overexploitation in India*. www.macrographics.com

Robert, M., Bergez, J. E., & Thomas, A. (2018). A stochastic dynamic programming approach to analyze adaptation to climate change – Application to groundwater irrigation in India. *European Journal of Operational Research*, *265*(3), 1033–1045. https://doi.org/10.1016/j.ejor.2017.08.029

Roy, A. D., & Shah, T. (2002). Socio-ecology of groundwater irrigation in India. *Intensive Use of Groundwater Challenges and Opportunities*, 307–335.

Sayre, S. S., & Taraz, V. (2019). Groundwater depletion in India: Social losses from costly well deepening. *Journal of Environmental Economics and Management*, *93*, 85–100. https://doi.org/10.1016/j.jeem.2018.11.002

Schrieks, T., Botzen, W. J. W., Wens, M., Haer, T., & Aerts, J. C. J. H. (2021). Integrating Behavioral Theories in Agent-Based Models for Agricultural Drought Risk Assessments. *Frontiers in Water*, *3*(September). https://doi.org/10.3389/frwa.2021.686329

Shah, T. (2009). Climate change and groundwater: India's opportunities for mitigation and adaptation. *Environmental Research Letters*, *4*(3). https://doi.org/10.1088/1748-9326/4/3/035005

Solomon, D. S., & Rao, N. (2018). Wells and well-being in South India: Gender dimensions of groundwater dependence. *Economic and Political Weekly*, *53*(17), 38–45.

Tierolf, L., Haer, T., Botzen, W. J. W., de Bruijn, J. A., Ton, M. J., Reimann, L., & Aerts, J. C. J. H. (2023). A coupled agent-based model for France for simulating adaptation and migration decisions under future coastal flood risk. *Scientific Reports*, *13*(1), 1–14. https://doi.org/10.1038/s41598-023-31351-y

Udmale, P., Ichikawa, Y., & Manandhar, S. (2014). International Journal of Disaster Risk Reduction Farmers ' perception of drought impacts , local adaptation and administrative mitigation measures in Maharashtra. *International Journal of Disaster Risk Reduction*, *10*, 250–269. https://doi.org/10.1016/j.ijdrr.2014.09.011

Udmale, P., Ichikawa, Y., Manandhar, S., Ishidaira, H., Kiem, A. S., Shaowei, N., & Panda, S. N. (2015). How did the 2012 drought affect rural livelihoods in vulnerable areas? Empirical evidence from India. *International Journal of Disaster Risk Reduction*, *13*, 454–469. https://doi.org/10.1016/j.ijdrr.2015.08.002